# Intramolecular chaperone-mediated secretion of an Rhs effector toxin by a type VI secretion system

Tong-Tong Pei[1,4], Hao Li[1,4], Xiaoye Liang[1,2,4], Zeng-Hang Wang[1], Guangfeng Liu[3], Li-Li Wu[1], Haeun Kim[2], Zhiping Xie [1], Ming Yu[1], Shuangjun Lin[1], Ping Xu [1] & Tao G. Dong [1,2✉]

Bacterial Rhs proteins containing toxic domains are often secreted by type VI secretion systems (T6SSs) through unclear mechanisms. Here, we show that the T6SS Rhs-family effector TseI of *Aeromonas dhakensis* is subject to self-cleavage at both the N- and the C-terminus, releasing the middle Rhs core and two VgrG-interacting domains (which we name VIRN and VIRC). VIRC is an endonuclease, and the immunity protein TsiI protects against VIRC toxicity through direct interaction. Proteolytic release of VIRC and VIRN is mediated, respectively, by an internal aspartic protease activity and by two conserved glutamic residues in the Rhs core. Mutations abolishing self-cleavage do not block secretion, but reduce TseI toxicity. Deletion of VIRN or the Rhs core abolishes secretion. TseI homologs from *Pseudomonas syringae*, *P. aeruginosa*, and *Vibrio parahaemolyticus* are also self-cleaved. VIRN and VIRC interact with protein VgrG1, while the Rhs core interacts with protein TecI. We propose that VIRN and the Rhs core act as T6SS intramolecular chaperones to facilitate toxin secretion and function.

[1] State Key Laboratory of Microbial Metabolism, Joint International Research Laboratory of Metabolic & Developmental Sciences, School of Life Sciences and Biotechnology, Shanghai Jiao Tong University, 200240 Shanghai, China. [2] Department of Ecosystem and Public Health, University of Calgary, 3330 Hospital Dr. NW, Calgary, AB T2N4Z6, Canada. [3] National Center for Protein Science Shanghai, Shanghai Advanced Research Institute, Chinese Academy of Sciences, 201204 Shanghai, China. [4]These authors contributed equally: Tong-Tong Pei, Hao Li, Xiaoye Liang. ✉email: tdong@ucalgary.ca

It is fascinating that some proteins manage to keep similar sequences and structural folding in both prokaryotic and eukaryotic cells despite billions of years of evolution[1–4]. One such example is the YD-repeat (tyrosine–aspartate) protein family whose members include many toxins in both Gram-positive and Gram-negative bacteria as well as eukaryotic teneurins that are conserved transmembrane adhesion proteins with critical functions in embryogenesis and neural development[5–9]. In bacteria, YD-repeat signatures are often found within the core domain of Rhs (rearrangement hot spot) proteins followed by a divergent C-terminal toxin domain; some Rhs also possess an additional N-terminal domain with unknown function[7,10–12]. Members of Rhs proteins include the antibacterial WapA in Gram-positive *Bacillus subtilis*[13], the ABC insecticidal Tc-toxins[7,14,15], and secreted effectors by the type VI secretion system (T6SS) in Gram-negative species[10,11]. Known T6SS Rhs effectors have been shown to be frequently associated with an N-terminal PAAR domain or downstream of VgrG and PAAR encoding genes[8,10,11,16–19].

The T6SS plays a critical role in interspecies interaction and during infection by translocating toxic effectors to bacterial and eukaryotic host cells[20–23]. The needle-like T6SS resembles a contractile bacteriophage-like tail, consisting of a VipA/B outer sheath, an Hcp inner tube, and a transmembrane-baseplate complex[24,25]. The tip of the tube is "sharpened" by a spike complex made of a VgrG trimer and a PAAR protein[26]. The inner Hcp tube carries effectors out of the cell upon sheath contraction. T6SS effectors exhibit diverse toxicities against essential cellular targets in eukaryotic[27,28] and prokaryotic cells[29–33]. Each antibacterial effector has a cognate immunity protein that confers self-protection[29,30,33]. Some effectors are Hcp, VgrG, and PAAR structural proteins with evolved C-terminal functional domains[18,19,26,28,33]. Non-structural effectors can be secreted by binding to the inner Hcp tube or to the tip VgrG/PAAR proteins[10,19,33]; the latter often involves chaperone proteins that are required for stabilization and delivery of effectors[34–38].

We previously predicted a chaperone-dependent T6SS effector TseI in *Aeromonas dhakensis*, a waterborne pathogen associated with skin and soft-tissue infection, gastroenteritis, and bacteremia[34,39,40]. TseI, made of 1545 amino acids, represents one of the largest T6SS effectors. Here we show that TseI belongs to the Rhs/YD-repeat family and possesses two self-cleavage sites, between residues C420 and P421 and between residues L1433 and S1434, resulting in three fragments, VIRN (VgrG-interacting Rhs N terminus), Rhs core, and VIRC (VgrG-interacting Rhs C-terminus). Importantly, the cleaved products remain in complex through noncovalent interactions. VIRC encodes an endonuclease that degrades DNA and confers the bacterium-killing toxicity of TseI. Secretion of TseI also requires upstream encoded VgrG1 and a conserved DUF4123-domain chaperone TecI[34]. The Rhs core but not the VIRN/C domains binds to the TecI chaperone. The VIRN domain and the Rhs core are required for VIRC delivery and co-translocated by T6SS, suggesting they may function as secreted chaperones for VIRC. Our results demonstrate the T6SS secretion of self-cleaved Rhs effectors, intramolecular chaperones, and add important new insights in understanding the physiological roles of widespread Rhs homologs in bacteria.

## Results

**TseI–TsiI as an effector–immunity pair**. The *tseI* gene is located in a multi-gene operon containing three upstream genes encoding two T6SS structural proteins Hcp1 and VgrG1, and a chaperone TecI (Fig. 1a). We also predicted a previously unannotated gene that we name *tsiI*, whose start codon overlaps with the last amino

acid codon and the stop codon of TseI. To determine the function of TseI, we first analyzed its protein sequence using Phyre2 and blastp[41,42]. TseI comprises three distinct regions, an N terminus of unknown function, a middle Tc-toxin domain, and a C-terminal Tox-HNH-EHHH domain (Fig. 1a). The Phyre-predicted Tc-toxin domain contains 838 residues with 100% confidence and 20% identity with TccC3 in *Photorhabus luminescens* (PDB entry 4O9X)[14]. The Tc-toxin domain also contains multiple Rhs/YD-repeat signatures. The C-terminal domain of TseI belongs to the predicted HNH Endonuclease VII toxin superfamily (pfam15657) with conserved [ED]H motif and two histidine residues (Supplementary Fig. 1).

To test if *tseI* and *tsiI* encode a T6SS-dependent effector–immunity pair[23,29], we constructed a deletion mutant, Δ*tseI*c*tsiI*, lacking the predicted *tseI* functional region including the toxin-coding sequence and the *tsiI* gene. Competition assay shows that the Δ*tseI*c*tsiI* mutant was efficiently outcompeted by wild type but not by the T6SS-null Δ*vasK* mutant (Fig. 1b). Survival of the Δ*tseI*c*tsiI* was restored when complemented with a plasmid-borne *tsiI* (Fig. 1b).

We next determined whether the C-terminal domain confers the observed toxicity. Sequence alignment shows that it contains several highly conserved predicted catalytic residues (Fig. 1a). We mutated two predicted catalytic sites, histidine 1497 and residues 1507–1509, histidine, phenylalanine, and histidine, to alanine residues (H1497A and HFH-AAA). Wild-type TseI and its two mutants were expressed using an arabinose-inducible vector pBAD in *E. coli*[43]. Survival of *E. coli* was severely reduced in wild-type samples compared with the mutants, indicating that H1497A and HFH-AAA abolished TseI toxicity (Fig. 1c). Western blot analysis confirmed that the nontoxic mutants were expressed (Supplementary Fig. 2A). In addition, these two corresponding chromosomal *tseI* mutants failed to outcompete the Δ*tseI*c*tsiI* mutant (Fig. 1d; Supplementary Fig. 2B).

We then sought to test whether the predicted HNH Endonuclease VII domain possesses endonuclease activities in vitro. To obtain sufficient amount of wild-type TseI and overcome its toxicity during expression, we co-expressed His-tagged TseI and untagged immunity TsiI and then purified TseI under denaturing conditions, followed by regeneration (Fig. 1e; Supplementary Fig. 2C). Nontoxic mutants H1497A and HFH-AAA were purified under the same condition. Results show that only wild-type TseI degraded plasmid DNA, confirming that the conserved H1497 and HFH (1507–1509) are critical residues for the endonuclease activity (Fig. 1e). We used H1497A and HFH-AAA interchangeably hereafter in this study.

The nontoxic H1497A mutation also enabled us to test the interaction of TseI and TsiI using the bacterial two-hybrid assay. We constructed C-terminal fusions of TseI H1497A and its immunity protein TsiI with the two split fragments T25 (224 amino acids) and T18 (175 amino acids) of the *Bordetella pertussis* adenylate cyclase (CyaA)[44], respectively. Results show that chimeric TseI and TsiI but not the T6SS transcriptional regulator VasH could reconstitute CyaA activity (Fig. 1f). Collectively, these results demonstrate that TseI is a T6SS endonuclease effector and TsiI is the cognate immunity protein that confers protection against TseI through direct interaction.

**TseI is cleaved into three fragments**. While purifying His-tagged wild type and mutant TseI, we noticed three cleaved products (Fig. 2a; Supplementary Figs. 2C and 3A). Using N-terminal amino acid sequencing, we determined that TseI is cleaved after residues cysteine 420 and leucine 1433 (Fig. 2a). These three resulting fragments correspond to the predicted Rhs sequence (116 kDa), the N-terminal domain (45 kDa), and the C-terminal

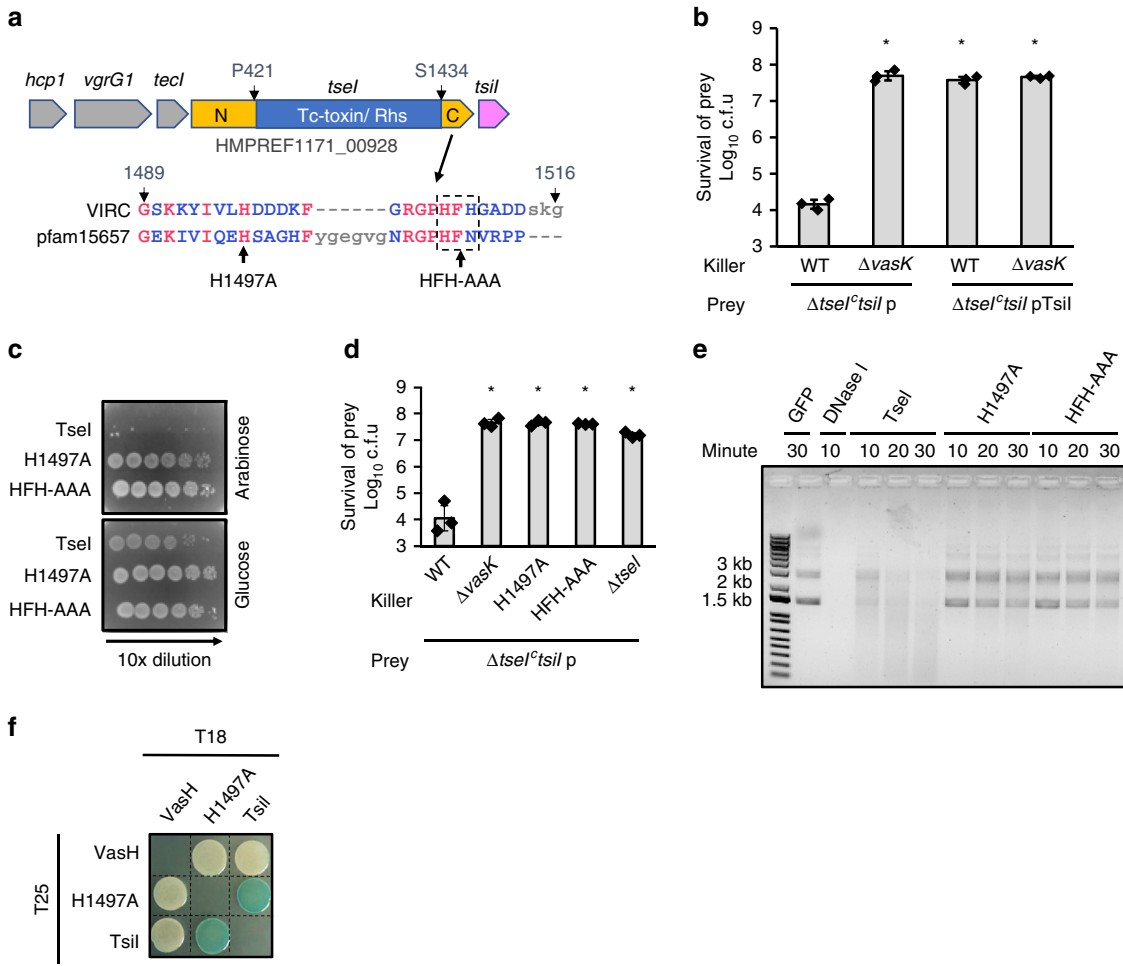

**Fig. 1 Characterization of TseI–TsiI effector–immunity pair. a** Operon structure and predicted catalytic residues of TseI. The TseI N-terminal domain VIRN and C-terminal VIRC domain are indicated as N and C for simplicity. The first residues for the middle Rhs domain and for the VIRC are indicated. The immunity gene *tsiI* is not annotated in the draft genome. Sequence of the VIRC toxin region was aligned with the consensus sequence of Pfam15657 that represents a conserved domain family of the predicted HNH/Endonuclease VII toxin with a characteristic conserved [ED]H motif and two histidine residues. **b** Competition assay of wild type (WT) and the T6SS-null Δ*vasK* mutant against the effector–immunity deletion mutant Δ*tseI*ᶜ*tsiI*. Survival of Δ*tseI*ᶜ*tsiI* complemented with an empty vector (p) or a vector carrying the immunity gene *tsiI* was quantified after co-incubation with the killer strains. **c** Toxicity of expressing TseI and its catalytically inactive mutants in *E. coli*. TseI and its mutants were expressed on pBAD vectors and survival of *E. coli* was tested by serial plating on arabinose (induction) and glucose (repression) plates with 10-fold dilutions. Expression of wild type and mutant TseI was confirmed by western blot analysis shown in Supplementary Fig. 2A. **d** Competition assay showing the activity loss of TseI mutants. Survival of killer and prey strains that carry pBAD vectors with different antibiotic resistance was enumerated by serial plating on selective medium for the killer and the prey, respectively. Survival of the killer strains is shown in Supplementary Fig. 2B. **e** DNA degradation by TseI and its mutants. Purified pUC19 plasmid was treated with GFP, DNase I, TseI, and two TseI catalytic mutants. DNA was sampled at different time points and examined by electrophoresis on an agarose gel. For activity assays, TseI proteins were purified under denaturing conditions as described in Methods and quality checked by SDS-PAGE analysis in Supplementary Fig. 2C. Green fluorescence proteins (GFP) was purified similarly except for without denaturing treatment and was used as a negative control. For each reaction, 0.3 μg protein was used. Commercial DNase I (1 unit) was used as a positive control. **f** Bacterial two-hybrid analysis of TseI–TsiI interaction. Proteins fused with the adenylate cyclase T25 or T18 subunits were co-expressed in the reporter strain BTH101 as indicated. Positive interaction results in color development on an LB-X-gal plate. A known T6SS transcriptional regulator VasH was used as a negative control. For killing assays (**b**, **d**), error bars indicate the mean ± standard deviation of three biological replicates and statistical significance was calculated using a two-tailed Student's *t*-test, *P < 0.01. Source data are provided as a Source Data file. Data in **b**–**f** are representative of at least two replications.

toxicity domain (observed size 18 kDa, predicted size 13 kDa, pI 9.57). We also analyzed each excised protein band using LC-MS analysis and the results confirmed the identity of each fragment (Supplementary Fig. 3A). Interestingly, the three cleaved fragments were co-eluted regardless of the His-tag position, suggesting they remain in complex post cleavage (Supplementary Fig. 3B).

**Mutating conserved residues blocks N- and C-terminal cleavage.** To understand the cleavage mechanism, we first used a

bioinformatics approach to identify conserved residues. Using blastp, we found over >1000 TseI highly similar homologs (E-value = 0, identity > 45%) in the NCBI non-redundant protein database, primarily in the two genera *Aeromonas* and *Pseudomonas* (Supplementary Fig. 4). To reduce redundancy, we selected a set of 48 representative sequences with one from each species from the top 1000 hits (Supplementary Data 1). Sequence alignment shows that a highly conserved signature "PVSMVTGEELL", located right after the N-terminal cleavage site and at the beginning of the Rhs core, is present in all 48

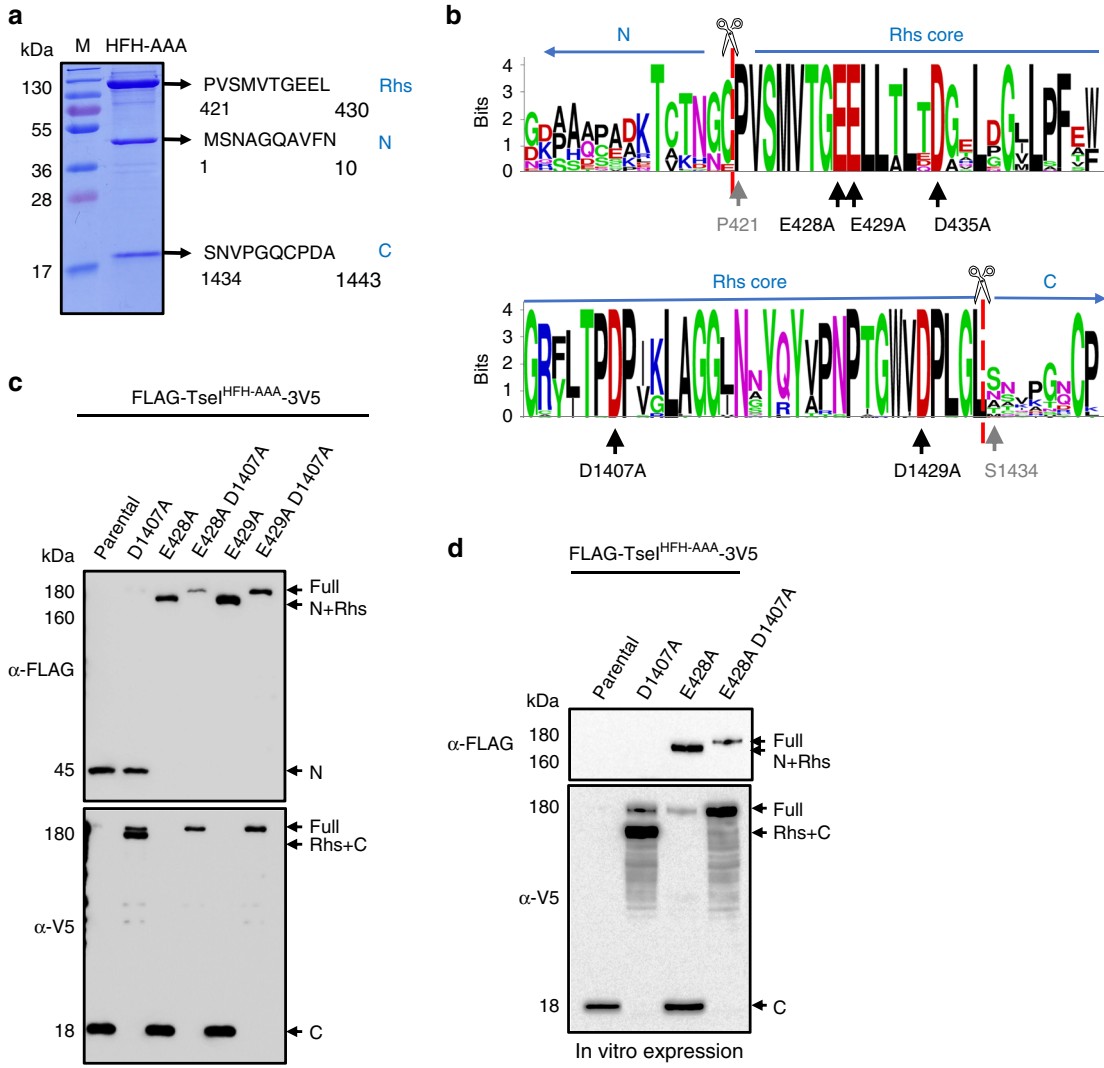

**Fig. 2 Characterization of TseI cleavage and key residues. a** Cleavage sites determined by N-terminal sequencing. Each band was excised for N-terminal Edman sequencing as well as LC-MS/MS identification (see also Supplementary Fig. 3A). **b** Weblogo depicting conserved residues of Rhs N-/C-terminal sequences deriving from sequence alignment of 48 representative Rhs homologs. Sequences are provided in Supplementary Data 1. Black arrows indicate the predicted key activity residues that are mutated in this study while gray arrows indicate the first residue of Rhs and VIRC post cleavage, respectively. **c** Western blotting analysis of TseI and its cleavage-defective mutants. All constructs were cloned to pETDUET1 vectors with an N-terminal FLAG tag and a C-terminal 3V5 tag. Proteins were induced in *E. coli* with 0.01 mM IPTG. The nontoxic HFH-AAA TseI mutant is used as the parental construct. The same pETDUET1 constructs were also used for in vitro expression shown in **d**. In vitro expression was performed with a PURExpress® In Vitro Protein Synthesis Kit following the manufacturer's instruction. Synthesized proteins were subject to SDS-PAGE analysis, followed by western blot analysis with anti-FLAG and anti-V5 antisera. Source data are provided as a Source Data file. Data in **a**, **c**, **d** are representative of at least two replications.

representative sequences (Fig. 2b). The C-terminal sequence also contains a conserved sequence "TPDPxxLAGGxNxYx-YxPNPTGWVDPLGL" right before the C-terminal cleavage site. Comparison of TseI with the known self-cleavable Tc-toxin TccC3 shows that the C-terminal cleavage sites are identical and the consensus signature resembles the internal aspartyl protease found in TccC3 (Fig. 2b; Supplementary Fig. 3C)[14].

To test the effect of these identified conserved residues on TseI cleavage, we constructed a series of single and combinatorial point mutations changing the two conserved C-terminal protease catalytic residues D1407 and D1429, and the conserved N-terminal residues E428, E429, and D435 to alanine in the nontoxic mutants HFH-AAA or H1497A backgrounds. We then examined the cleavage of those mutants expressed in cells by western blot analysis as well as purified proteins by SDS-PAGE analysis (Fig. 2c; Supplementary Fig. 5A, B). Results collectively show that mutations D1407A and D1429A abolished C-terminal

cleavage while E428A and E429A but not D435A abolished N-terminal cleavage. Double mutations of D1407A with E428A or E429A resulted in full-length non-cleaved TseI (Fig. 2c). To eliminate the remote possibility that the C-terminal endonuclease inactivation by the HFH-AAA mutation is responsible for cleavage, we also constructed cleavage mutants of wild-type TseI and expressed them in the ΔtseI mutant in which the chromosomal-encoded immunity TsiI confers protection from TseI toxicity. Western blot analysis of expressed proteins shows results consistent with the mutant cleavage phenotypes (Supplementary Fig. 5C, D).

Next, we tested whether TseI cleavage is dependent on any external protease. We used a commercial in vitro protein synthesis kit with defined purified components to express TseI variants flanked by N-FLAG and C-3V5 epitope tags. Western blot analysis shows that TseI was cleaved similarly in vitro, and mutations of D1407A and E428A also abolished cleavage,

suggesting that TseI is self-cleaved (Fig. 2d). Using the anti-FLAG antibody, we did not detect a distinct band corresponding to the cleaved N terminus or full-length TseI in parental and D1407A samples. Instead, we found multiple nonspecific signals across all samples which likely result from lower stability of TseI N terminus under the in vitro expression condition (Fig. 2c, d and source data).

Notably, although N-terminal cleavage is still detectable in the C-terminal cleavage-defective mutants, the amount of full-length TseI was substantially enriched in comparison with wild type (Fig. 2c, d; Supplementary Fig. 5A, C), suggesting that the C-terminal protease-inactivating mutations reduced the efficiency of N-terminal cleavage.

**Cleavage is critical for TseI-mediated competition**. We next tested if self-cleavage is important for T6SS-dependent delivery of TseI. Using the immunity-defective mutant ΔtseI$^c$tsiI as prey and the ΔtseI mutant complemented with plasmid-borne TseI and its N- or C-cleavage-defective mutants as killer, we found that cleavage is critical for the killing of ΔtseI$^c$tsiI (Fig. 3a). Control experiments testing protein expression and survival of killer strains ruled out the possibilities that the increased prey survival is due to poor expression of those plasmid-borne constructs (Supplementary Fig. 6A) or impaired growth of cleavage-defective killers during the competition assays (Supplementary Fig. 6B). In fact, despite that the E429A mutant killer showed a log more growth than the wild type or other mutant killer strains (Supplementary Fig. 6B), it was still impaired in killing the ΔtseI$^c$tsiI prey.

To provide further evidence, we also constructed chromosomal mutants defective in cleavage, D1407A, D1429A, E428A, and E429A. Competition assays against the ΔtseI$^c$tsiI prey confirm that cleavage is critical for TseI-mediated cell-to-cell competition (Fig. 3b). There was no difference in the survival of killer strains (Supplementary Fig. 6C).

**Non-cleaved TseI mutants are secreted to the extracellular medium**. Next, we tested the effect of cleavage on TseI secretion by expressing C-terminal 3V5-tagged wild type and cleavage-defective mutant TseI in the ΔtseI and the T6SS-null ΔvasK mutants. Using antisera to the C-terminal V5 tag, Rhs, and N terminus respectively, we detected full-length TseI and N-cleaved TseI (Rhs+C) in the secreted samples of the D1407A and the D1429A mutants, while we detected the cleaved C and the Rhs fragments in wild-type samples (Fig. 3c). Secretion of N was ambiguous using the custom α-N antibody due to nonspecific signals but was later confirmed using a plasmid-borne 3V5-tagged N (Supplementary Fig. 6D). Similarly, T6SS-dependent secretion of the cleaved C terminus was detected in all samples expressing wild-type TseI, the E429A, and the D435A mutant TseI (Fig. 3d). Secretion of the N terminus and the Rhs was found in wild type and D435A samples while secretion of the N+Rhs fragment was detected in the E429A mutant sample. Full-length non-cleaved TseI mutants carrying double mutations of E429A D1407A or D435A D1407A were also secreted (Supplementary Fig. 6E). Detection of Hcp, the inner tube of the T6SS, and RpoB, the RNA polymerase subunit B, serves as indicators for T6SS functions and cell lysis, respectively.

To eliminate the possibility that plasmid-borne expression might have unexpected effects on TseI secretion, we also tested the effect of chromosomal mutations D1407A, D1429A, and constructed the ΔRL40 mutant by deleting the whole C-terminal internal protease lacking 40 amino acids from arginine 1394 to leucine 1433, respectively. Using the antiserum to the middle Rhs domain, we detected secretion of full-length TseI and the

fragment corresponding to TseI lacking the N terminus (Rhs+C) in mutants D1407A and D1429A, further supporting that full-length TseI is secreted regardless of cleavage (Fig. 3e). Surprisingly, the ΔRL40 deletion abolished not only C-terminal cleavage but also secretion of the mutant TseI and reduced the amount of the Rhs+C fragment (Fig. 3e). The latter two effects might result from the loss of protease activities or indirectly from nonspecific effects of deletion. To confirm that the internal C-terminal protease is not required for N-terminal cleavage, we performed SDS-PAGE analysis of purified ΔRL40 TseI$^{HFH-AAA}$ in comparison with TseI$^{HFH-AAA}$. Results show that the ΔRL40 mutant seemed to be less stable than its parental but the Rhs+C fragment lacking N was readily detectable in the ΔRL40, indicative of N-terminal cleavage (Supplementary Fig. 6F).

**Cleavage is important for TseI toxicity**. To understand how non-cleaved TseI mutants were secreted but severely impaired in outcompeting the ΔtseI$^c$tsiI mutant, we next tested whether cleavage affects TseI toxicity by comparing the survival of E. coli expressing arabinose-inducible wild type and cleavage-defective TseI mutants (Fig. 3f). The relative survival of cells between induced and uninduced conditions show that mutations D1407A and E429A attenuated toxicity in comparison with wild-type TseI, although both TseI D1407A and E429A mutants also exhibited moderate toxicities in comparison with the nontoxic HFH-AAA mutant (Fig. 3f). Considering that the physiological level of T6SS-delivered TseI is likely much lower than that of intracellular induction, such attenuated toxicity might account for the impaired killing of prey cell by cleavage mutants during competition.

**TseI secretion requires VgrG1 and chaperone TecI through direct interaction**. The genes upstream of tseI encode VgrG1 and a chaperone protein TecI (Fig. 1a). To test if TseI secretion is dependent on VgrG1 and TecI, we made deletion mutants of these two genes. Both mutants failed to secrete TseI or outcompete the ΔtseI$^c$tsiI mutant, suggesting that VgrG1 and TecI are required for TseI secretion (Fig. 4a, b; Supplementary Fig. 7A). We then tested if VgrG1 and TecI directly interact with TseI. Pull-down analysis shows that VgrG1 could pull down both TecI and TseI separately or together (Fig. 4c). Because TseI was detected against its C-terminal 3V5 tag, we could only detect its cleaved C-terminus but not full-length TseI probably due to efficient self-cleavage. As protein purification assay shows that the cleaved products remain in complex (Supplementary Fig. 3B), we then tested the interaction of each TseI fragment with VgrG1 and TecI, respectively. VgrG1 could interact with the N-terminal and C-terminal of TseI but not the middle Rhs core (Fig. 4d). We thus name the N and C terminus VIRN and VIRC, respectively. In contrast, TecI was found to interact with the middle Rhs core and full-length TseI but not the VIRN or the VIRC of TseI (Fig. 4e). Interaction of TecI with the full-length TseI might result from the intramolecular interaction of TseI domains since both VIRN (N) and Rhs (M) fragments can interact with the VIRC (C) toxin directly (Fig. 4f, g). This is consistent with the observation that all three fragments of TseI were co-eluted when either an N-terminal or a C-terminal 6His-tag was used (Supplementary Fig. 3B).

**VIRN and Rhs are required for toxin secretion**. Next, we tested whether the N-terminal VIRN and the Rhs domains are important for the secretion of the C-terminal VIRC toxin by expressing plasmid-borne full-length TseI, the N terminus deletion mutant (MC), and the VIRC domain (C) only mutant in the ΔtseI and the T6SS-null ΔvasK mutants. Western blot analysis of whole cell and secreted samples revealed that MC was cleaved but not secreted

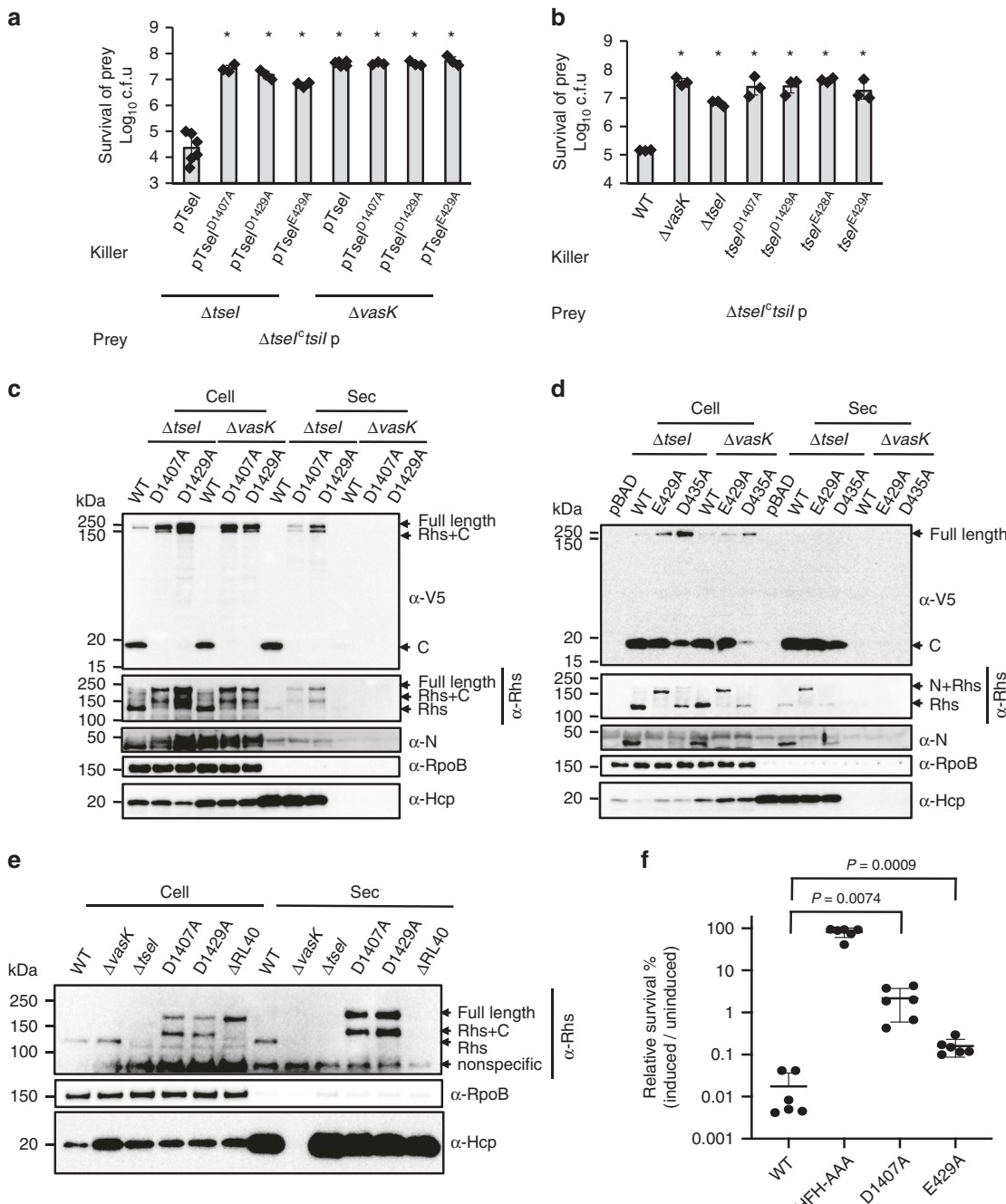

**Fig. 3 Effects of self-cleavage on TseI functions. a** Competition analysis of the Δ*tseI* mutant complemented with different TseI cleavage mutants. Killer strains expressing pBAD-TseI constructs are indicated and the prey strain is the Δ*tseI*ᶜ*tsiI* mutant. The Δ*vasK* mutant serves as a T6SS-null control. **b** Competition analysis of chromosomal *tseI* mutants against the Δ*tseI*ᶜ*tsiI* prey. For **a** and **b**, error bars indicate the mean ± standard deviation of at least three biological replicates (*n* = 6 for Δ*tseI* and Δ*vasK* carrying pTseI plasmid as killer, and *n* = 3 for the others) and statistical significance was calculated using a two-tailed Student's *t*-test, **P* < 0.01. **c** Secretion analysis of TseI C-terminal cleavage-defective mutants. **d** Secretion analysis of TseI N-terminal cleavage-defective mutants. For **c** and **d**, the Δ*tseI* mutant and the Δ*vasK* mutant hosting pBAD vectors expressing C-terminal 3V5-tagged proteins were induced with 0.01% arabinose. Protein expression and secretion was detected by western blotting analysis using antiserum to V5 for the C-terminus and full length and custom antisera for the middle Rhs fragment and the N terminus. **e** Secretion analysis of chromosomal mutants of TseI. The ΔRL40 mutant lacks the protease sequence from arginine 1394 to leucine 1433. TseI was detected using the antibody to Rhs. **f** Survival of *E. coli* expressing plasmid-borne wild-type TseI and its cleavage-defective mutants. All TseI proteins were cloned on pBAD vectors. Cells were treated with 0.2% glucose (repression) or induced with 0.01% arabinose for 2 h, after which *E. coli* cells were 10-fold serial diluted and plated on LB media containing 0.2% glucose to repress expression. Relative survival was calculated as the percentage of survived *E. coli* under induction versus repression conditions. Error bars indicate the mean ± standard deviation of six biological replicates. Statistical significance was calculated using a two-tailed Student's *t*-test. The RNA polymerase subunit RpoB serves as a control for cytosolic expression and cell lysis, and the T6SS inner tube Hcp serves as a positive control for T6SS delivery in **c**, **d** and **e**. Source data are provided as a Source Data file. Data in **a–f** are representative of at least two replications.

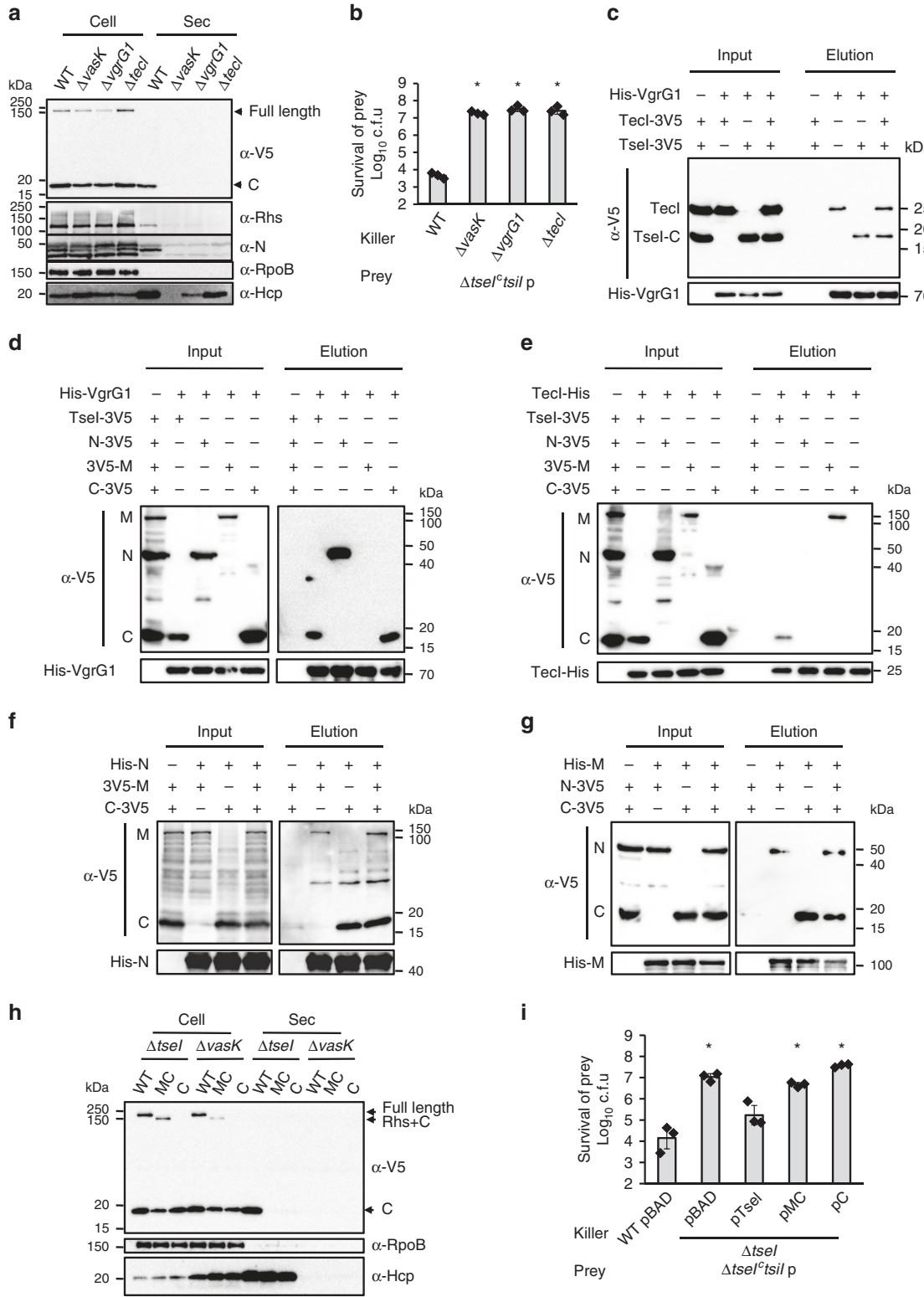

suggesting VIRN is required for TseI secretion but not for its cleavage (Fig. 4h). VIRC alone was not secreted either (Fig. 4h). Consistent with this result, only full-length TseI but not its truncated mutants could functionally complement the Δ*tseI* mutant in a competition assay against the Δ*tseI*^c*tsiI* immunity defective mutant (Fig. 4i; Supplementary Fig. 7B). Notably, the VIRN and the Rhs domain are not required for toxicity since intercellularly expressed MC and C were toxic to *E. coli* (Supplementary Fig. 7C). Unlike the VIRN domain (Supplementary

Fig. 6D), Rhs was not secreted when it was expressed alone (Supplementary Fig. 7D). Expression of the nontoxic H1497A VIRC domain alone was barely detected in the cell lysate suggesting VIRC is unstable and prone to degradation or insoluble in the absence of VIRN and Rhs (Supplementary Fig. 7E).

**N- and C-terminal cleavage of TseI homologs.** Because Rhs/YD-repeat family contains over 120,000 members, we next asked

**Fig. 4 TseI secretion requires VgrG1 and chaperone TecI. a** Secretion analysis of TseI in the Δ*vgrG1* and the Δ*tecI* mutants. Wild type and deletion mutants expressing pBAD-TseI-3V5 were induced with 0.01% arabinose. Expression was detected using α-V5 antibody. **b** Competition analysis of Δ*vgrG1* and Δ*tecI*. Killer strains are indicated and prey strain is the Δ*tseIᶜtsiI* mutant. **c** Pull-down analysis of VgrG1 with TecI and TseI. Full-length TseI^HFH-AAA mutant and VIRC^HFH-AAA mutant were used in all pull-down analyses to avoid toxicity. Because C-terminal 3V5-tagged TseI was used in **c–e**, we could only detect the cleaved C-terminus but not full-length TseI due to self-cleavage. All pull-down analyses were performed by mixing cell lysates of individually expressed proteins. **d** Pull-down analysis of VgrG1 with TseI domains. The Rhs fragment is indicated as M (middle) for simplicity in **d–g**. **e** Pull-down analysis of TecI with TseI domains. **f** Pull-down analysis of TseI N terminus with its M and C fragments. **g** Pull-down analysis of TseI-M with its N and C fragments. **h** Secretion analysis of truncated TseI. MC refers to the mutant TseI lacking the N-terminal domain, while C refers to the C-terminal toxin domain only. TseI and its mutants were tagged with a 3V5 C-terminal tag and expressed on pBAD vectors. Plasmids were transformed to the Δ*tseI* and the Δ*vasK* mutants as indicated. For **a** and **h**, the RNA polymerase subunit RpoB serves as a control for cytosolic expression and cell lysis, and the T6SS inner tube Hcp serves as a positive control for T6SS delivery. **i** Competition assay of the Δ*tseI* mutant complemented with truncated TseI against the Δ*tseIᶜtsiI* prey. For **b** and **i**, error bars indicate the mean ± standard deviation of three biological replicates and statistical significance was calculated using a two-tailed Student's *t*-test, *P < 0.01. Source data are provided as a Source Data file. Data in **a–i** are representative of at least two replications.

whether other Rhs homologs are also subject to N- or C-terminal cleavage. We first used sequence alignment to compare TseI with several published homologs including the Tc-toxin TccC3, T6SS Rhs effectors in *Dickeya dadantii*, *Pseudomonas aeruginosa*, *Serratia marcescens*, *Vibrio parahaemolyticus*, WapA in *Bacillus subtilis*[7,10,11,45,46], as well as newly predicted ones in *Pseudomonas syringae* and *Vibrio aerogenes* (Supplementary Data 2). TseI is clustered with *P. syringae* homolog PSPTO_5438 and divergent from the previously published Rhs effectors (Fig. 5a). Domain analysis shows that TseI homologs are featured with a conserved Rhs core flanked by divergent N- and C-terminal sequences (Supplementary Fig. 8). The N terminus sequences show variable lengths, with some possessing the T6SS-associated PAAR motif[26]. The N terminus cleavage site including the residues that affect cleavage is conserved in some but not all homologs while the C-terminal cleavage-activity residues are highly conserved except for RhsA in *Escherichia coli* strain EDL933 (Fig. 5b).

We next tested the cleavage of selected homologs, *P. syringae* PSPTO_5438, *P. aeruginosa* PA2684, and *V. parahaemolyticus* VP1517. Expression of PSPTO_5438 was highly toxic to *E. coli* unless its downstream gene PSPTO_5439 was co-expressed, indicative of an effector–immunity pair (Fig. 5c). Purified PSPTO_5438 exhibited three bands corresponding to the predicted size of cleaved N- and C-terminal products (Fig. 5d; Supplementary Fig. 9A). Mutating the N-terminal conserved residue E412 to alanine reduced the N-terminal cleavage and the E413A mutation abolished it (Fig. 5d). The D1385A and D1407A mutations of the C-terminal protease abolished C-terminal cleavage (Fig. 5d). Lastly, we tested expression of PA2684 in the cytosol and in the periplasm using a twin-arginine signal (Tat) since it was previously proposed that PA2684 targets the periplasm[10]. We found that PA2684 was also cleaved releasing a C-terminal fragment corresponding to the predicted cleavage size regardless of its localization (Fig. 5e). Similar cleavage and toxicity were also observed for VP1517 (Fig. 5f; Supplementary Fig. 9B–D).

## Discussion

Rhs proteins (InterPro IPR001826) form a large ancient protein family with members existing in all kingdoms. Although T6SS-dependent Rhs effectors have been previously reported[10,11,26,47,48], we show for the first time self-cleaved Rhs effectors that are secreted by the T6SS. The Rhs effector TseI is self-cleaved into three interacting fragments, the N-terminal VIRN domain, the conserved Rhs core, and the C-terminal VIRC toxin domain (Fig. 6). Blocking cleavage has little effect on effector secretion but impairs toxicity and effector-mediated competition. We demonstrate that the N terminus and the C-terminal toxin of TseI interact with VgrG1, and the Rhs middle core interacts with the chaperone TecI (Fig. 6). Unlike the C-terminal toxin, neither the

Rhs core or the N terminus possess any toxic domain nor exhibit any toxicity when cytosolically expressed. In addition, even the nontoxic C-terminus mutant could not be easily expressed and purified in comparison with the Rhs core and the N terminus due to instability or poor solubility (Supplementary Fig. 7E). Because the biological function of TseI is dependent on the C-terminal toxin whose secretion requires not only VgrG1, chaperone TecI, but also the N-terminal domain and the Rhs core, we propose that the Rhs core and the N terminus function as intramolecular chaperones for the C-terminal toxin.

Secreted T6SS chaperones have not been previously reported. Notably, some of the type VII secretion system (T7SS) secreted substrates display chaperone functions[49]. For example, the *Mycobacterium* T7SS secretes heterodimeric substrates EsxAB, of which EsxA displays membrane pore formation activities while its binding partner EsxB prevents EsxA aggregation[50,51]. Although it remains unclear why TseI requires cleavage for its cell-to-cell competition, the VIRC toxin might require VIRN and Rhs core for stability at the physiological levels delivered to recipient cells. Importantly, we found that VIRN could interact with the VIRC toxin in the absence of Rhs, suggesting the chaperoning role of VIRN and Rhs could be synergistic.

Toxins play a key role in bacterial pathogenesis and are likely under continuous selection through evolution. It is possible that the large number of Rhs-associated toxins serve as a reservoir accessible by delivery systems[8,9]. By contrast to the internal aspartic protease-mediated C-terminal cleavage, the mechanism of N-terminal cleavage is unclear and structural prediction fails to identify any protease signature. We identified two key glutamic acid residues for N-terminal cleavage. Although we cannot determine whether they are catalytic residues of a novel protease fold in this study, identification of these key residues enables us to construct full-length non-cleaved mutants of TseI and its homologs, thereby paving the way for structural characterization that would in turn reveal key information about the cleavage mechanism, as well as the overall structural arrangement of Rhs effectors.

Known structures of Rhs/YD-repeat homologs, including the bacterial Tc-toxin C-subunit TcC and the eukaryotic teneurin extracellular domain, show that the Rhs/YD-repeat core assembles a beta-barrel shell[7,14,15,52–54]; the key structural difference between them is the position of the C-terminus (Supplementary Fig. 10A). While the C-terminus of teneurin protrudes out of its shell[52,53], the Tc-toxin TcC shell and the TcB subunit encapsulate the C-terminal toxin that is released through the TcA-formed translocation channel when the holotoxin enters target cells[7,15,55]. In comparison, expression of wild type and cleavage-defective TseI is toxic to cells suggesting the C-terminal toxin can reach its DNA substrate. Therefore, despite the predicted similarity of TseI to Tc-toxin TcC, TseI might not encapsulate its C-terminal VIRC nuclease toxin. This is further supported by the bacterial two-

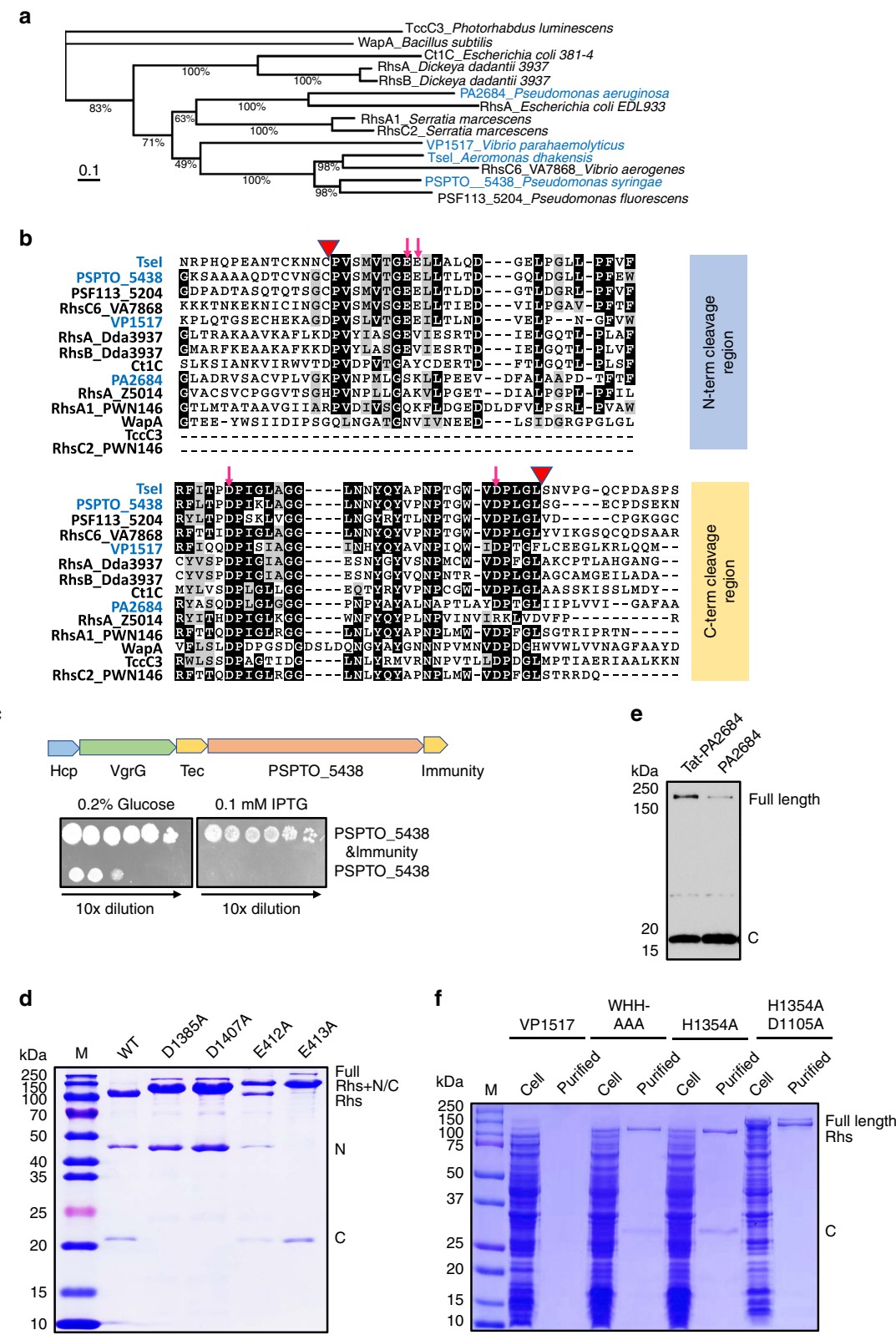

hybrid assay that showed direct interaction of non-cleaved TseI mutants, the protease-inactive D1407A-D1429A mutant and the protease deletion ΔRL40 mutant, with the immunity protein TsiI when they were fused to the split fragments of CyaA (Supplementary Fig. 10B). These results suggest the arrangement of Rhs-VIRC complex likely resembles that of the eukaryotic teneurin extracellular domain. This prediction warrants validation by structural characterization of TseI in future studies.

TseI self-cleavage is also reminiscent of the common auto-processing of proteases. Some proteases, e.g. human procathepsin L and carboxypeptidase Y, are synthesized as an inactive pre-cursor to prevent from unwanted toxicity and activated upon

**Fig. 5 Cleavage of TseI homologs. a** Maximum-likelihood phylogeny of select sequences including previously published Rhs proteins. Phylogeny was constructed using IQ-TREE web server with bootstrap 1000 times and values indicated. Proteins experimentally tested in this study are highlighted in blue in **a** and **b**. **b** Alignment of the N-terminal and the C-terminal cleavage regions of Rhs proteins. Red triangle indicates cleavage site and red arrow indicates mutation sites that abolish cleavage. **c** Operon structure and toxicity of PSPTO_5438. PSPTO_5438, and its predicted immunity gene were cloned to pET28a. Survival of *E. coli* BL21(DE3) expressing PSPTO_5438 alone or together with immunity was compared on plates containing glucose (repression) or IPTG (induction) with 10-fold serial dilutions. **d** Critical residues that abolish PSPTO_5438 cleavage. N-terminal His-tagged PSPTO_5438 and mutant proteins were co-expressed with untagged immunity proteins, purified with nickel columns, and compared by SDS-PAGE analysis. Mutations D1385A and D1407A inhibited cleavage of the C-terminal while E413A inhibited N-terminal cleavage, resulting in enriched accumulation of Rhs+C and Rhs+N fragments. E412A attenuated but did not fully block N-terminal cleavage. **e** C-terminal cleavage of PA2684 detected by western blotting analysis. PA2684 was expressed with a C-terminal FLAG tag in the periplasm (Tat-signal added) and in the cytoplasm of *E. coli*, respectively. C-terminal cleavage was detected in both samples. **f** Comparison of purified VP1517 and its mutants. C-terminal His-tagged VP1517 and its mutants were expressed and purified with a nickel column. WHH-AAA refers to three amino acid mutations of tryptophan, histidine, and histidine (from 1328 to 1330) to alanine that inactivate the toxicity of VP1517. Mutation H1354A also abolished toxicity. Mutation D1105A inhibited the C-terminal cleavage of the VP1517 H1354A mutant. Cleavage of these proteins was confirmed by western blotting analysis in Supplementary Fig. 9D. Source data are provided as a Source Data file. Data in **c–f** are representative of at least two replications.

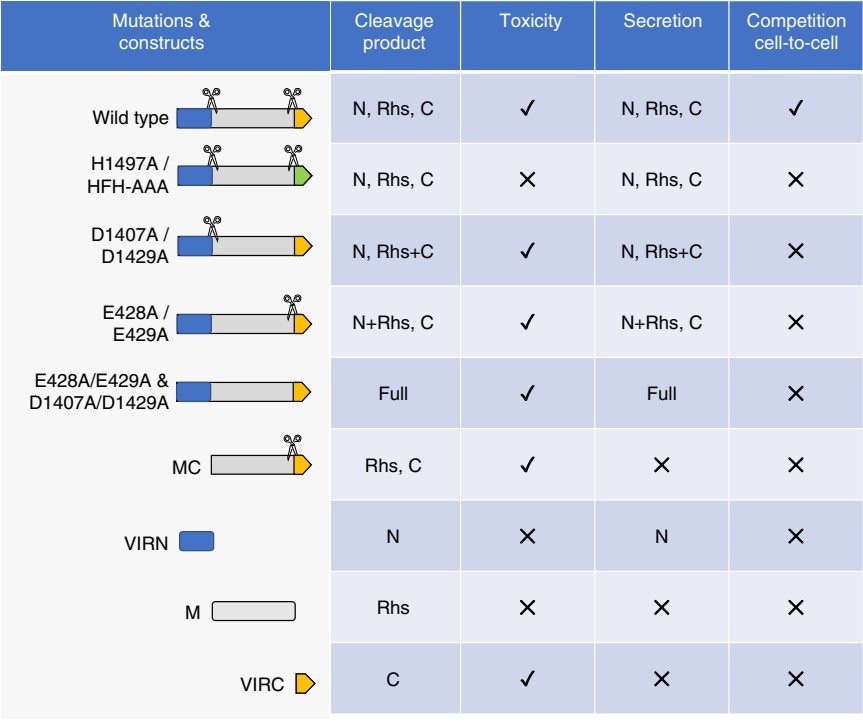

| Mutations & constructs | Cleavage product | Toxicity | Secretion | Competition cell-to-cell |
|---|---|---|---|---|
| Wild type | N, Rhs, C | ✓ | N, Rhs, C | ✓ |
| H1497A / HFH-AAA | N, Rhs, C | ✗ | N, Rhs, C | ✗ |
| D1407A / D1429A | N, Rhs+C | ✓ | N, Rhs+C | ✗ |
| E428A / E429A | N+Rhs, C | ✓ | N+Rhs, C | ✗ |
| E428A/E429A & D1407A/D1429A | Full | ✓ | Full | ✗ |
| MC | Rhs, C | ✓ | ✗ | ✗ |
| VIRN | N | ✗ | N | ✗ |
| M | Rhs | ✗ | ✗ | ✗ |
| VIRC | C | ✓ | ✗ | ✗ |

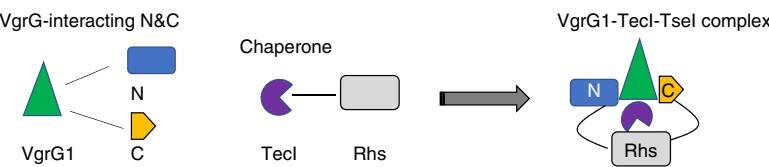

**Fig. 6 A schematic model of intramolecular chaperone-mediated secretion of TseI.** TseI mutations and their resulting effects are summarized. TseI is self-cleaved into three fragments. Both N- and C-terminal domains interact with VgrG, named VIRN and VIRC respectively, while the Rhs core (M) interacts with the Tec chaperone protein. It is likely that VgrG1, TecI, and TseI form a transient protein complex prior to delivery. Note that VIRN, Rhs, and VIRC likely maintain noncovalent interaction even after cleavage. VIRC secretion requires VIRN and Rhs that are also co-secreted by the T6SS. Therefore, VIRN and Rhs may serve as secreted intramolecular chaperones to facilitate VIRC functions.

specific environmental signals by cleaving off the inhibitory peptide[56–58]. Other proteases including subtilisin and alpha-lytic protease employ self-cleaved peptides to guide the folding and stability of the protease domain[59–61]. These cleaved fragments are named intramolecular chaperones as they are encoded in the primary gene sequences[62,63]. The self-cleaved VIRN and Rhs core thus distinguish themselves from the known T6SS chaperones[34,35] and represent a new class of intramolecular chaperone for toxin secretion. These findings, together with the

self-cleavage of Rhs proteins as T6SS effectors, further highlight the versatility of T6SS delivery[38].

## Methods
**Bacterial strains and growth conditions.** Strains, plasmids, and primers used in this study are listed in Supplementary Table 1 and Supplementary Data 4 and 5. Cultures were grown in LB ([w/v] 1% tryptone, 0.5% yeast extract, 0.5% NaCl) aerobically at 37 °C unless otherwise stated. Antibiotics were used at the following concentrations: streptomycin (100 μg/ml), ampicillin (100 μg/ml), kanamycin (50

µg/ml), irgasan (25 µg/ml), gentamicin (20 µg/ml), chloramphenicol (25 µg/ml for *E. coli*, 2.5 µg/ml for SSU).

**Protein secretion assay.** Cultures were grown aerobically in LB with appropriate antibiotics at 30 °C to $OD_{600} = 1$ and collected by centrifugation at $2500 \times g$ for 8 min. Pellets were resuspended in fresh LB and induced at 30 °C for 1 h. Expression of genes on pBAD vectors was induced with 0.01% [w/v] L-arabinose. Cells were centrifuged at $2500 \times g$ for 8 min at room temperature. Pellets were resuspended in SDS-loading dye and used as whole-cell samples. Supernatants were centrifuged again and then precipitated in 20% [v/v] TCA (trichloroacetic acid) at −20 °C for 20 min. Samples were centrifuged at $15,000 \times g$ for 30 min at 4 °C and pellets were washed twice with acetone, air-dried, and resuspended in SDS-loading dye. Whole cell and secretion samples were boiled for 10 min before SDS-PAGE analysis.

**Western blotting analysis.** Proteins were resolved on an SDS-PAGE gel and transferred to a PVDF membrane (Bio-Rad) by electrophoresis. The membrane was blocked with 5% [w/v] non-fat milk in Tris-buffered saline with Tween-20 (TBST) buffer (50 mM Tris, 150 mM NaCl, 0.05% [v/v] Tween-20, pH 7.6) for 1 h at room temperature, incubated sequentially with primary antibodies and secondary HRP-conjugated antibodies in TBST with 1% [w/v] milk. Signals were detected using the Clarity ECL solution (Bio-Rad). Monoclonal antibodies to epitope tags were purchased from Sigma Aldrich (FLAG, Product # F1804 and 6His, Product # SAB4600386), Thermo Scientific (V5, Product # 37-7500), and Biolegend (RpoB, Product # 663905). RpoB, the beta subunit of RNA polymerase, was used as a control for equal loading and cell lysis in western blotting analysis. The polyclonal antibodies to Hcp, the VIRN, and the Rhs domains of TseI were custom-made by Shanghai Youlong Biotech (Supplementary Fig. 11). The secondary antibodies (anti-mouse or anti-rabbit IgG HRP linked) were purchased from Cell Signaling Technology (CST, Product # 7076S and # 7074S, respectively). RpoB and secondary antibodies were used at 1:20,000 dilution, while others at 1:10,000 dilution.

**Bacterial competition assay.** Killer and prey strains were grown in liquid culture to exponential phase ($OD_{600} = 1$) and stationary phase ($OD_{600} = 2$) respectively. Cells were centrifuged at $4500 \times g$ for 3 min, resuspended in fresh LB, mixed together at a ratio of 5:1 (killer:prey) and spotted on LB-agar plates. After co-incubation for 3 h at 37 °C, cells were resuspended in 500 µl LB and a series of 10-fold dilutions were plated on LB plates with antibiotics. The mean $Log_{10}$ c.f.u. of recovered preys was plotted and error bars show mean ± standard deviation between three biological replicates. A two-tailed Student's *t*-test was used to determine *p* values.

**Protein expression and purification in bacteria.** Genes of interest were cloned into pETDUET1, pET28a, or pET22b vectors that were transformed to *E. coli* BL21 (DE3). Cells were grown in LB with appropriate antibiotics to $OD_{600}$ ~0.6 at 37 °C.

Protein expression was induced with 1 mM IPTG at 20 °C for 18 h. The cells were centrifuged at $10,000 \times g$ for 30 min. The pellets were resuspended in lysis buffer (50 mM $NaH_2PO_4$ pH 8.0, 300 mM NaCl, 10 mM imidazole) and lysed by sonication. Lysates were centrifuged at $10,000 \times g$ for 40 min and the supernatants were transferred onto Ni-NTA or Cobalt-NTA resin (Smart-lifesciences). Proteins were eluted in elution buffer (50 mM $NaH_2PO_4$ pH 8.0, 300 mM NaCl, and variable concentrations of imidazole). Eluted samples were analyzed by SDS-PAGE and western blotting.

In vitro protein expression was performed with a PURExpress® In Vitro Protein Synthesis Kit (NEB) following the instruction of the manufacturer.

For TseI purification under denaturing conditions, His-tagged proteins were first purified with Ni-NTA resin, eluted with elution buffer A (50 mM $NaH_2PO_4$ pH 8.0, 300 mM NaCl, 250 mM imidazole), and dialyzed with dialysis buffer (50 mM $NaH_2PO_4$ pH 8.0, 300 mM NaCl) three times to remove imidazole. Protein samples were then mixed with the denature buffer (50 mM $NaH_2PO_4$ pH 8.0, 300 mM NaCl, 8 M guanidine hydrochloride) to a final concentration of 6 M guanidine hydrochloride, incubated at room temperature for 1 h, and mixed with Ni-NTA resin. Proteins were then eluted with elution buffer B (50 mM $NaH_2PO_4$ pH 8.0, 300 mM NaCl, 6 M guanidine hydrochloride, 250 mM imidazole) and dialyzed in a series of buffers with a decreasing gradient of guanidine hydrochloride at 4M, 2M, 1M, and no guanidine hydrochloride, respectively. All dialysis steps were performed at 4 °C.

**Protein pull-down assays.** Genes were cloned into pBAD, pETDUET1, and pET22b vectors for expression with either His or 3V5 tags. Cultures were grown in 10 ml LB with appropriate antibiotics to $OD_{600}$ of 0.6, and induced with 0.1% [w/v] L-arabinose at 30 °C for 3 h or 1 mM IPTG overnight at 20 °C. Cells were centrifuged at 4500 r.p.m. for 10 min and resuspended in 1 ml of lysis buffer (20 mM Tris pH 8.0, 500 mM NaCl, 50 mM imidazole with protease inhibitor (Thermo Scientific)). Resuspended cells were sonicated (20 × 5 s) and cell debris was removed by centrifugation ($15,000 \times g$ for 15 min). Clarified supernatants were mixed and loaded to Ni-NTA resin (Smart-lifesciences), washed five times with wash buffer (20 mM Tris pH 8.0, 500 mM NaCl, 50 mM imidazole), and eluted in

100 µl elution buffer (20 mM Tris pH 8.0, 500 mM NaCl, 500 mM imidazole). Cell lysates and eluted samples were analyzed by western blotting.

**Bacterial two-hybrid assay.** Proteins of interest were fused to the T18 and T25 split domains of the *Bordetella* adenylate cyclase as previously described[34,44]. The two plasmids encoding the fusion proteins were co-transformed into the reporter strain BTH101. Three independent colonies for each transformation were inoculated into 500 µl of LB medium. After 5 h growth at 30 °C, 5 µl of each culture were spotted onto LB plates supplemented with ampicillin, kanamycin, IPTG, and X-Gal and incubated for 10 h at 30 °C and then 10 h at room temperature. The experiments were done in triplicate and a representative result is shown.

**Bioinformatics analysis and homology modeling.** All gene sequences of *A. dhakensis* SSU are retrieved from the draft genome assembly (GenBank NZ_JH815591.1). The gene sequences of *hcp1-vgrG1-tecI-tseI-tsiI* are provided in Supplementary Data 3. Benchling was used to manage and analyze DNA and protein sequences and to predict *tsiI* as a putative open reading frame[64]. TseI sequence was analyzed with Phyre2 and blastp to identify homologs and species distribution[41,42]. A representative set of 48 sequences were manually selected from the top 1000 hits and downloaded from the NCBI database. Homologs were aligned using Clustal Omega[65]. The resulting alignment was used to generate a WebLogo[66]. Structural models of TseI were constructed using Phyre2 (ref. [41]) and the resultant models were compared with known structures teneurin and Tc-toxin Tcc3 using Chimera[67]. Maximum-likelihood phylogeny was constructed using IQ-TREE web server with bootstrap 1000 times[68]. The iTOL server was used to visualize the phylogenetic tree[69].

**Protein toxicity assay.** Cells expressing different plasmid constructs were grown in LB supplemented with 0.2% [w/v] glucose at 37 °C overnight. Cells were then collected and resuspended in fresh LB and grown to $OD_{600} = 1$. A series of 10-fold dilutions were plated on LB plates containing 0.01% [w/v] L-arabinose (for pBAD vectors), 0.1 mM IPTG (for pET vectors), or 0.2% [w/v] glucose, respectively. Each experiment was repeated three times, with one representative experiment shown.

**N-terminal Edman sequencing and LC-MS/MS analysis.** Purified TseI was resolved on an SDS-PAGE gel and protein bands were excised individually. The sequence of the first 10 amino acids for each band was determined by Edman sequencing and performed at the BiotechPack Scientific company. LC-MS/MS analysis of excised bands was performed at the Southern Alberta Mass Spectrometry core facility. Original data will be provided upon request.

**Reporting summary.** Further information on research design is available in the Nature Research Reporting Summary linked to this article.

## Data availability
The source data underlying Figs. 1B, 1D, 2C–D, 3A–F, 4A–I, 5E and Supplementary Figs. 2B, 6B–E, 7A–B, and 7D are provided as a Source Data file. Other data supporting the findings of this study are available within the paper or from the corresponding author upon request.

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

## Acknowledgements

This work was supported by funding from National Key R&D Program of China (2018YFA0901200), National Natural Science Foundation of China (31770082), the Canadian Institutes of Health Research, and the Natural Sciences and Engineering Research Council of Canada. We thank Le Tang, Brianne Burkinshaw, and Steve Hersch for technical assistance. We thank Dr. Jun Zheng for sharing strains and data on protein VP1517. We thank Laurent Brechenmacher and the Southern Alberta Mass Spectrometry Centre for LC-MS/MS analysis. The funders had no role in study design, data collection and interpretation, or the decision to submit the work for publication.

## Author contributions

T.G.D. conceived the project. X.L. identified the *tseI-tsiI* pair and made the observation of TseI cleavage. T.-T.P., H.L., X.L., Z.-H.W., G.L., L.-L.W., and H.K. performed research; P.X., S.L., Z.X., and M.Y. provided key reagents and materials; T.G.D. wrote the manuscript with assistance from T.-T.P. and X.L.

## Competing interests

The authors declare no competing interests.
