## [Peer Review File · Nature Communications]

Reviewers' comments:

Reviewer #1 (Remarks to the Author):

In this work, Pei et al. report a comprehensive biochemical characterization of a recently discovered T6SS effector, TseI from *Aeromonas dhakensis*. TseI is composed of a central RHS core, an N-terminal domain that interacts with VgrG and a C-terminal cytotoxic domain. Using a combination of site-directed mutagenesis, protein secretion assays, bacterial competition experiments, bacterial two-hybrid assays and pulldown assays, the authors addressed and answered several key questions that are crucial for the function of TseI. After identification of the active site of TseI and showing that it forms an immunity pair with TsiI, two autoproteolytic cleavage events were identified: both the N-terminal domain and the C-terminal domain are cleaved off from the large central RHS domain. While the cleavage of the C-terminus is conserved in other RHS toxins, the cleavage of the N-terminus is a unique feature of TseI. Next, the authors show that only the N-terminal domain and the toxic C-terminal domain are secreted, while the central RHS domain remains in *A. dhakensis*, and that secretion depends on the presence of both VgrG and TecI. Finally, the authors identified homologs of TseI in *Pseudomonas syringae* and *P. aeruginosa*, that both also show cleavage of the N-terminus and C-terminus. The biochemical experiments were executed in an appropriate manner, and important controls were included. The work presented here describes the biological function of a T6SS-dependent RHS toxin and reveals some important novel findings, such as the cleavage of the N-terminal domain. I believe the article is suitable for publication in *Nature Communications*. However, there are some major issues that should be addressed before publication.

1. Introduction: A proper introduction to Tc toxins is missing. In particular the authors completely omit the work of the Raunser lab and do not cite it, although they use their atomic model of TccC3.
2. C-terminal toxin is not encapsulated: Since TseI is a homologue to TccC3 from *Photobacterium luminescens*, the cocoon or beta-barrel shell, as the authors name it, misses the B component, i.e. half of the barrel. Therefore, it is not surprising that the C-terminal toxin is not encapsulated. The SAXS experiment is vague, poorly done and illustrated and should be removed from the manuscript. The extension into which the C-terminus is modeled (Fig. 2i) could also be a part of the RHS cocoon or the N-terminal domain. The conclusion that the C-terminus is outside of the RHS cocoon is solely based on this SAXS experiment and should therefore also be removed from the manuscript. In addition, it does not make sense that the toxin is oriented in this direction. How can it remain bound to TseI after autoproteolysis if it is reaching out that far? This opens another important question that the authors can probably answer in a follow-up study: How can the C-terminal toxin remain attached to TseI after cleavage without an encapsulating barrel? A future structure of the protein would of course answer this question. In general, this study would greatly benefit from structural studies (such as X-ray crystallography or electron microscopy), that shed light onto the molecular organization of participating components, resulting in a more sophisticated model than currently presented in Fig. 7.
3. Discussion: "However, the Tc-toxin C-terminus has not been imaged inside the beta-barrel shell." This is not true. The authors should read the literature in detail and cite the appropriate papers, accordingly. A good start would be: *Annu Rev Microbiol.* 2019 Sep 8;73:247-265
4. Figure legends do not provide enough detail.
5. Figure 7 is not conclusive. The authors should invest more time to develop a nice figure that should be properly described and discussed in the main text and in the figure legend.
6. Data presentation in general: The authors created a large number of TseI variants to address several particular features of TseI, such as cleavage of N- and C-terminus, toxic activity and export, by site-directed mutagenesis. This results in many different mutants whose effects are described by an identical set of experiments that are presented in several figures, therefore it is very difficult for the reader to follow. An overview figure in which all mutations and their respective location is shown would be helpful. In addition, it would also be helpful if one important

experiment, such as the inter-species competition experiment (Fig. 1B ,D, Fig. 2D, Fig. 3B, Fig. 5B), would be presented as one concluding figure together with the mutant overview and the scheme in Fig. 7.

In the following, I provide comments and suggestions to improve the paper in a point-by-point manner. There are some minor language issues that should also be changed.

Abstract:

Please write clearly that TseI is cleaved two times, releasing the C-terminal toxin and the N-terminal VIRN.

Introduction:

p.3, l.5f.: Please re-write the sentence to make it clear that teneurins are only present in eukaryotes (animal kingdom).

p.3, l.16f: I suggest to write bacteriophage-like tail instead of bacteriophage tail; and trans-membrane without the hyphen. If you decide to write "The tip is sharpened", I suggest to write "sharpened" in "".

p.3, l.21: Please explain what is meant with "structural effectors". Modification of the cytoskeleton?

p.4, l.4f: Please stress the double cleavage, like suggested for the abstract, and please describe clearly where and when cleavage occurs.

p.4, l.10f: Here you refer to TccC3 without a citation, and later in the manuscript (e.g. p.10, l.17) you refer to TccC3 and cite Busby et al., 2013, who describe the structure of the TcB-TcC subunit of Yen-Tc. The correct citation for the structure of TcdB2-TcCC3 (*Phototribadus luminescens*), pdb ID 4O9X, is Meusch et al., 2014 (doi: 10.1038/nature13015).

Results:

p.4, l.19: Please explain your notation of tsiI.

p.5, l.3ff: It would be helpful to briefly explain the background of T6SS immunity pairs in this chapter.

p.5., l.18f: "highly toxic": Can you provide a comparison of toxicity with other T6SS effectors or other bacterial toxins?

p.6, l.13: "secreted medium" does not make sense.

p.6, l.20f: Deletion of the whole Asp protease domain (39 residues) might result in a different orientation of the toxin domain. Can you exclude this possibility?

p.7, l.7: Two recent publications show the encapsulation of the toxin domain in Tc toxins: Gatsogiannis et al, 2018 (doi: 10.1038/s41586-018-0556-6), Roderer et al, 2019 (doi: 10.1073/pnas.1909821116)

p.7, l.10ff: The Phyre models look like the two templates, indicating that TseI might look completely different. Therefore, these models are not really reliable, and I would show them in the SI and not as a main figure. I suggest to perform electron microscopy (negative staining of the purified protein is enough to get an idea about the shape) to get an additional proof that TseI looks like expected.

p.7, l.25f.: The fact that secretion is not possible in trans, i.e. that it is only possible when the complex is pre-formed is an important finding. Does it mean that the complex made up of VIRN, the RHS-domains and the toxin domain cannot be reconstituted from the individual domains? Please provide this information.

p.8, l.21: Why should the secretion of the C-terminus be influenced by the N-terminal cleavage?

p.9, l.13: What is meant with TseI-inactive mutants? Cleavage-inactive?

p.10, l.2ff: The pull-down results described here contradict the previous observations that TseI forms a non-covalent complex of all three parts. Can you provide an explanation? Does VgrG1 displace the VIRN and the RHS domain from the C-terminus of TseI?

p.10, l.13: You introduce the rationale of the term VIRN domain here, but the expression is already used in the introduction. In the first paragraphs of the results section, you do not use VIRN, but refer to the domain as N-terminal. This can be confusing, therefore I recommend to introduce the notion VIRN in the introduction and to use it consistently throughout the manuscript. You

furthermore showed that also the C-terminus interacts with VgrG1 (Fig.5D), which could then be called VIRC accordingly.

p.11, l.6ff: For PSPTO_5438, the toxic WT can apparently be expressed and purified (Fig.6D), whereas this is not easily possible for TseI. Please discuss why this is the case.

Discussion:

p.11. l.22: Teneurin citations are used to describe Tc toxins, please change (see above).

p12., l.7: To which inactive variants do you refer to (HFH-AAA)? With the toxic WT, I assume the cells die. What happens when the non-toxic variant is expressed (insoluble expression?) Please clarify.

p.12, l.13: Please describe briefly what EsxAB is.

p.13, l.3ff: Is it only a common motif of TseI and protease self-cleavage or is there also cleavage site conservation or a common mechanism?

An additional section of the discussion, in which you compare the size of the RHS domain of TseI with those of Tc and Teneurins, would be informative. Is the TseI RHS domain possibly too small to encapsulate the C-terminus?

Figures:

General remark: The figures are complex and almost every figure shows many different experiments. Therefore, it would be helpful for the reader to provide a brief description of the outcome of every experiment in the figure legend, especially in Figs. 2,3,5. In addition, a brief heading for every figure panel would help the reader to recognize which experiment is applied to address which point.

Fig. 1: It would be helpful to provide the residue numbering in the tseI scheme in Fig.1A, and to draw the respective domains in the real scale to each other. What is pfam15657, please write in the legend.

Fig.2 C,E: In both cases, variant D1407 is shown. Is it the very same experiment and two times shown in two different contexts?

Fig.1C and Fig. 2G: Is there a reason why the X- and Y-axis were changed in the two figure panels that show a bacterial-2-hybrid experiment?

Fig.2A,B: Is TseI-AAA in panel A the "parental" in panel B? Please name the variants consistently.

Fig. 2H,I: The Phyre-models look like the templates, therefore their significance is not very high. Which one was used in panel I? I suggest to put the phyre models to the SI and instead show a negative stain EM micrograph (or, more sophisticated, 2D classes of negatively stained TseI particles) that shows the shape of the molecule. To localize the N- and C-terminus of TseI in negative staining, immunogold labeling would be helpful.

Fig.3: Please provide a scheme of the variants MC and CT. In panel D, the non-specific signal that co-localizes with TseI-N is unfortunate and impairs the presence or absence of secreted protein, especially for pD435A. The context of the blot in panel E (T6SS-dependent secretion of the N-terminus) is difficult to understand, please provide a short description in the legend.

Is there a specific reason why variants E428A and E429A were used in different experiments?

In panel F, you refer to cleavage mutants, but you only show one mutant.

Fig.3 H: What should the right panel (anti-flag) show us? I can either see degradation of the in vitro expressed sample or unspecific binding.

Fig.4: It would be helpful to provide the target organism as a heading to the respective figure panels. For C, images of HEK cells treated with the different variants that potentially show differences in morphology would be informative (e.g. as a SI figure panel). What antibodies were used in B and C?

Fig.5 A: Are vgrG1 and tecI deletion mutant strains? Please clarify.

Fig. 5D: Why is the N fragment alone binding to VgrG, but not when full length TseI is provided? In this case, I would expect also binding of the N fragment after self-cleavage (or at least binding of the N-M fragment).

Fig. 5E: A more detailed explanation would help the reader to understand why the C-terminal fragment is pulled down by TecI when full length TseI is provided as input.

Fig.5 F: The experimental result suggests that the N-terminal domain directly interacts with the C-terminal domain without the central RHS domain. Please discuss this unexpected finding. Furthermore, the indication of "N" is missing.

Fig. 5G: The signal for N- and C-fragment in the elution control due to unspecific binding is almost as strong as the actual signal in the other elutions.

Fig.6: Please indicate TseI clearly in A and B, and also the other three homologs that were expressed in this work. Please also provide the names of the organisms. In B, one of the Asp in "RhsA_Z5014" is not conserved, which should result in cleavage deficiency. If the authors also have access to this particular gene, it would be a great addition together with the other three homologs. Is there a particular reason why PSPTO_5438 is shown in Fig. 6, and VP1517 is shown in Fig. 8? It would be easier to read if both data were in one figure.

Fig.7: This figure needs to be improved, both, on an illustrative and content-related view. The scheme should be more refined, both with respect to protein structures and cell shape. What is the reason to draw a direct interaction of N and C only in the target cell? Information from pulldown and secretion assays with TsiI would help to establish a more complete picture of what is going on and could be ultimately implemented into Fig. 7.

Supplementary information:

SI Fig.2C: How does the alignment continue after the cleavage site? Please add more residues to visualize.

SI Fig.3: Do all fits of the SAXS data result in the same molecular shape presented in Fig. 2?

SI Fig. 5: The legend refers to Fig. 3F, but no in vitro expression is shown in this panel.

SI Fig. 6: Please provide the names of the organisms. The size of the respective RHS domains could also be provided (see my comment regarding the discussion above).

Reviewer #2 (Remarks to the Author):

In this manuscript, Pei et al characterize an *Aeromonas* T6SS effector/immunity pair named TseI/TsiI. The authors nicely describe a self-cleavage mechanism that releases the C-terminal toxin domain from the central Rhs region. They further discover an N-terminal domain named VIRN that is also cleaved off the Rhs, and is even secreted. It is suggested to function as an intramolecular chaperone. The concluding results suggest that these self-cleavage mechanisms are common in Rhs toxins.

This is an interesting study that will be of interest to the microbiology community, and especially to those interested in secreted bacterial toxins. There are some experiments that I think can be improved with additional controls to support the conclusions, and they are listed below. The section on possible activity of the effector in eukaryotic cells, however, is not properly supported by the experiments and should be either removed or improved.

Major comments:

1) Much of this work relies on TseI secretion experiments, yet the authors did not show that the TseI secretion that they are detecting is indeed T6SS-dependent (to be clear, I am not questioning that TseI is a T6SS-delivered effector, as this was properly shown in competition assays). The authors only test T6SS-dependent secretion for the N-terminal part in Fig. 3E. A secretion assay of TseI from WT vs. T6SS-mutant should be shown for the full-length protein (show the T6-mediated secretion of C- and N-terminal in the context of a full length effector).

2) Fig. 3A (p. 7 lines 22-23): Can the authors explain why the CT is secreted in *tseI*-deletion when expressed from a plasmid (far right lane), and also why is it not secreted then from WT? Since CT is expressed at a lower level in the cell compared to the full length TseI (as seen in the "Cell" lanes on the left), this secreted CT appears to be quite a significant amount. This result either contradicts the authors hypothesis, or it is the result of cell lysis/non-T6SS-dependent secretion (which refers back to my previous comment on the lack of T6SS-dependent secretion experiment).

3) P.9 lines 4-6 (and Fig. 3F-G): since the CT was delivered even in the E429A mutant, but no complementation was seen during competition, the authors should show (immunoblot) that the mutant was expressed in the donor strain during the competition shown in 3F. Also, some explanation should be provided here (or in the discussion) about to the contradicting results shown in these panels. Is N-terminal cleavage required for prey intoxication/delivery or not? The question is left unaddressed.

4) Fig. 4: The experiment shown in A is not necessarily testing for effector toxicity in yeast, but rather for stability of the plasmid. To directly show toxicity in yeast, the authors should clone the effector into a yeast inducible vector (e.g. under galactose-inducible promoter) and show that the expression of the effector is toxic. The current experiment possibly shows that the effector, being a DNase, is interfering with the transformation or with the stability of the plasmid (since the authors test for the presence of the plasmid that allows for growth in absence of uracil).

5) To the best of my knowledge, no antibacterial T6SS DNase effector has been shown to perform as a trans-kingdom toxin. If the authors are making such a claim here, they need to provide further experimental support, such as visualizing effect on yeast/HEK293 DNA under a fluorescence microscope, or purifying DNA from cells expressing the effector and showing that it is degraded *in vivo* upon effector expression.

Minor comments:

1) P.5, lines 7-9 (Fig. 1C): Expression of TseI is toxic in bacteria (shown in Fig. 1D). How was it expressed in the BACTH strain without immunity (with VasH instead of TsiI)? Same question is relevant in Fig. 2G.

2) It makes more sense that the detailed explanation on BACTH (p.6 line 24 – p.7 line 4) will appear when Fig. 1C is discussed.

3) p. 7 lines 1-12 (Fig. 2H): Since the N- and C-terminal domains were not modeled, this panel seems irrelevant and can be removed from the manuscript.

4) p.8 lines 14-15: since now there are 2 cleavage sites, this sentence is confusing. I suggest to rephrase it so that the reader clearly understands which cleavage site is mentioned.

5) The numbers in the X-axis of Fig. 3C (Weblogo) are illegible.

6) Fig. 3D: please denote the non-specific signal from the N-terminal antibody with arrow or asterisk.

7) Supplemental Fig. 4: while D345A retains N-terminal cleavage, it appears that all the tested mutations were hampered in C-terminal cleavage, since the lower band seen in the 1 TseIHFA-AAA lane is not present in any other lane. Can the authors comment on that?

8) Fig. 3D: can the authors comment as to why the secretion of TseIN is significantly hampered in the D435A mutant, even though N-terminal cleavage seems unaffected?

9) Fig. 3F: why is a qualitative (n=1) spotting assay shown for competition now when all previous experiments were shown as quantitative (n=3) bar graphs? Was this not also performed in triplicates with 3 repetitions?

10) Fig. 3G: some controls are missing to show that the signal is T6SS-dependent. For example, show that no signal is detected when the pTseI plasmids are in a T6SS mutant.

11) Fig. 3H:

a) Why is E428A used instead of the E429A that was used in all previous assays?

b) It is not clear what is shown in the anti-FLAG panel. There are many bands and it is difficult to determine which band is relevant.

c) It seems that in the E428A, the same pattern as in the parental lane is visible even though there should be no N-terminal cleavage (Supplemental Fig. 4). Can the authors please comment as to why they are seeing (what appears to be) N-terminal cleavage in the E428A mutant in this assay?

12) Fig. 5C (and P.10 lines 304): it would be helpful if the authors add size markers to this panel to allow the reader to see that only the TseI-C is being pulled down by VgrG1.

13) Fig. 5G: in the elution panel, double clicking on the figure in adobe reader colors the figure in blue, and reveals significant bands detected even in the lane without His-M. Is this just the result of the reduced quality of the PDF figure?

14) P.10 lines 19-20: *Vibrio parahaemolyticus* RHS effector vp1517 had been previously predicted to be a T6SS effector (Salomon et al, Plos Pathogens, 2015).

15) P. 12 lines 6-7: "In addition, the C-terminal toxin domain, active or inactive, could only be purified when co-expressed with the Rhs core and VIRN" – what figure does this refer to? Was this attempted directly?

16) P. 12 lines 15-16: "... the toxin might require VIRN for stability when delivered to recipient cells..." - since the possible function of VIRN as a chaperone is one of the central conclusions in this work, perhaps the authors should test expression of the C-terminal toxin domain in *E. coli* to determine if it is stable and toxic on its own, and if co-expression of VIRN affects the stability or activity of the toxin domain? That would explain why VIRN is also secreted. On the same topic, the model in Fig. 7 suggests that N and C interact after delivery to the prey cell, but no experimental support for this is provided.

Reviewer #3 (Remarks to the Author):

Intramolecular chaperone-mediated secretion of Rhs/YD-repeat effector toxin by the type VI secretion system

Comments to the authors:

In this paper the authors investigate the function and delivery mechanism of an Rhs homolog from *Aeromonas dhakensis*. The authors find that this effector (known as TseI) is mobilized by T6S and delivered to other *Aeromonas dhakensis* bacteria where it inhibits the growth of bacteria lacking cognate immunity. The toxic effect of the C-terminal toxin is confirmed to be a DNase and an adjacent gene provides immunity by direct binding to the C-terminal toxin. The authors further characterize the protein and find that it is subjected to autocleavage (similar to secreted Rhs

toxins (Tc toxins) in the C-terminus as well as the N-terminus. The authors show that the three fragments remain bound to one another by immunoprecipitation. Surprisingly, they show that the C-terminal toxin is not encapsulated within the Rhs-core as suggested previously, but that it is cleaved off and attached to the outside of the Rhs-core. The Rhs-core is not secreted but remains in the delivering cell, whereas the N-terminus and C-terminal toxin are secreted. The authors propose a model where the Rhs-core, the N-terminal fragment of Tse1 and TEC acts as chaperons that facilitate delivery of the Tse1 C-terminus to neighboring cells. If correct, this is a very interesting and previously unrecognized aspect of Rhs protein homologs and would be very interesting to the field. However, significant amount of detail is lacking from the manuscript. For example, there are no descriptions of how the plasmids in the study were generated, or what parts of the protein antibodies were raised against. No raw data is provided for the figures and a number of important controls are lacking. This raises a number of concerns listed below which need to be addressed in order to make this manuscript suitable for publication.

Main concerns:

1) The conclusion that the toxin must be exposed within the cell based on the fact that transformation of a plasmid with Tse1 results in toxicity in bacteria, yeast and HEC283T cells without immunity. This is not the only reason that these constructs could give toxicity. There could be internal expression of the C-terminal toxin from an internal translation start site within the C-terminus that results in toxicity. The same is true for the yeast two hybrid screen. Thus, the toxicity data do not prove that the C-terminus is exposed and toxic while bound to the Rhs-core. The authors would need to check if there is internal expression of their toxin by for example introducing stop codons within the Tse1 ORF before the DPXGL motif in order to show that the C-terminus is only toxic when expressed from the full-length ORF.

2) The authors show that the non-cleavable E429A still secretes C-terminal toxin, but is unable to inhibit target cells. Yet, in Fig 3G it seems as the toxin is indeed delivered to target cells. This is very contradictory and should be commented upon, how do the authors suggest this would work mechanistically? Why would the toxin not be toxic in the receiving cell? However, many controls are missing for these experiments (listed below), which might clarify the picture.

a. Fig 3G. The HiBiT/LgBiT assay is not convincing. How do you know that LgBiT is actually delivered to the cells and not to the extracellular milieu? The color could come from lysing targets releasing HiBiT to the supernatant. Since you have a functioning split cAMP assay, you could repeat the experiment with these strains and show that actual intact individual cells receive the toxin. Or, LgBiT fused to a protein known to be secreted to the extra-cellular milieu could work as a negative control. The pMC control is not valid as it is not delivered outside of the cell. At a minimum a control showing the lack of cell lysis in the assay should be included.

b. Fig 3D. An appropriate control i.e. a protein that is secreted should be added to the analysis.

c. Fig 3D. What is the extra band in the N-terminal blot? This is not present in Figure S9 where the antibodies are verified, could you please expand on this? Why is the Rhs core so much less abundant than the NT and CT? If these are indeed cleavage products, would you not expect a ratio of 1:1:1. Could this be the reason why it is not detected in the secreted fraction? Also, the mutants seem to make even less Rhs core, could this be the reason that there is less inhibition?

d. Fig 3E Cf_u values for the inhibitor should be shown in addition to the target. Otherwise it isn't possible to assess how expression of the different Tse1 variants affect inhibitor fitness. If inhibitor fitness is affected, this could be the reason that you do not see any inhibition in the competition.

3) Over-expression in the eukaryotic cells to show toxicity does not really add to the paper, unless you could also show that the toxin is actually delivered to these cells. Could you do competitions with yeast/HEC 283T cells?

4) Although it is likely that the protein is a nuclease considering the HNH nuclease domain, more evidence or controls are needed to prove this. The loading lane has been cut from the gel in Fig 1A and there are smears in the TseI samples. It is possible that TseI binds to the DNA and gets stuck in the loading lane, therefore it is important to show the lane. The samples could also be run on a protein gel and stained for DNA. Or, actual DNase activity could be shown in bacteriophage over-expression or receiving toxin by DNA staining and microscopy.

5) The secreted fraction on western blots in figures 2E and 3D are not very clean. In figures 2E and 3D in particular, other components of the Tse1 protein (except the C-terminus) can be identified in the secreted fraction. Yet, the authors claim that only the C-terminus and N-terminus are secreted. In addition, an appropriate control i.e. a protein that is secreted should be added to the analysis.

6) The authors claim that mutation of D1407A and D1429A do not affect N-terminal cleavage, but on the SDS gel it is obvious that the N-terminal fragment in these mutants is less abundant (Fig 2B). In addition, two additional larger bands appear, which look as they would correspond to full-length and full-length – NT, where most of the protein seems to be full-length though. Could the authors comment on this?

7) A full annotated genome of the *Aeromonas* strain is not available on NCBI, only shotgun sequencing. This makes it difficult to evaluate the genetic mutations created in the study. Could the authors please provide accession no:s of the contigs, gene ID:s for the genes relevant in this study.

8) Raw data is not provided for the experiments. Please provide raw data for all experiments.

9) Cfu values for the inhibitor should be shown in addition to the target. Alternatively, immunity protection controls for each competition could suffice as well. Otherwise it isn't possible to assess how expression of the different Tse1 variants or mutations of the T6S components affect inhibitor fitness. If inhibitor fitness is affected, this could be the reason that you do not see any inhibition in some competitions.

10) From fig s7 it seems as the effectors lack PAAR domains? If so, this should be commented on in the text.

Figure 5. There is a large inconsistency in the background seen when using the anti-V5 antibody. In some of the blots like 5D and 5E there are almost no unspecific binding and in others like 5F, the background is massive. Why is that?

Figure 4. The authors use a transformation assay into yeast/HEC293T cells to say that Tse1 is toxic also in eukaryotic cells. However, there is no control for the transformation of wild type Tse1, only negative data. It could just be that this plasmid was not taken up by the yeast/HEC293T cells. A regulatable promoter should be used instead so that the same transformation can be plated/grown on inducing /repressing conditions in order to prove that it is expression per se that is toxic.

Minor comments:

1. Page 5, line 13. "Expression of TseI and its mutants, H1497A and HFH-AAA, in *E. coli* showed that survival of cells was reduced by 5-logs when expressing TseI in comparison with cells

expressing tseI mutants (Figure 1D), indicating that the mutations abolish toxicity." Unclear and indirect, please rephrase.

2. Page 6, line 12. "We found that only wild type TseI C-terminal fragment was detected in the secreted medium suggesting full length wild type TseI and its two non-cleavage mutants D1407A and D1429A are not secreted (Figure 2C)." The sentence lacks a comma after medium and that after suggesting. In addition, Tse1 and its mutants is not grammatically correct, please rephrase to: "wild type and mutant Tse1" throughout the paper. This is also a very bold statement considering that there are additional bands in many of the blots (see above).

3. Page 8 Line 19. Please give aa numbers to the EED residues mutated.

4. Page 8, line 20. Do you mean Fig S5 instead of or in addition to S4?

5. Add size estimations for the westernblots in Figs 3D, 3E, 3H5B-G. Also, when blots are cropped for the main figures the entire gels should be shown in supplement.

6. Figure 4B-C. What antibody was used to detect Tse1 in yeast and HEC293T cells?

7. Figure 4B. Since the D1407A mutation results in Rhs-core + CT and NT fragments and not full-length Tse1, it should say "core" on the left of the blot and no "full" as this is confusing.

8. Figure 4C. Can full length Tse1 be detected in the HEC293T cells? If not, is it possible that only the C-terminal toxin is produced from another promoter within the ORF?

9. Figure S2. Why do you get less of the N-terminal fragment when purifying with an N-terminal his-tag?

10. Figure legends are not adequate in explaining what the figure shows. Remove all the interpretation of the data from the figure legend (e.g Figure S4 and S8). The authors should be allowed to interpret the data themselves. Please provide more details about what is shown in the figure instead. See examples of lacking detail below (this is not a comprehensive list).

a. All figures: What organisms were used for the different experiments? Sometimes this is stated, sometimes not.

b. Figure 1D and 2F. What dilutions were plated?

c. Figure 2B. Purified proteins – how were the proteins purified?

d. Figure 4B-C. What antibodies were used?

e. Figure S4 and S7. What does 40mM and 250mM stand for?

f. Figure S9. Where is the data with the Δ tse1 mutant?

Many thanks to all reviewers for your helpful feedback and ideas. After performing the suggested experiments using additional controls and protein-specific antibodies, we have made substantial revisions about the mode of TseI secretion that not only help us improve our premature model but also strengthen the conclusions. We have also combined and rearranged N-/C-cleavage sections and the related figures and removed non-essential data (toxicity in yeast and mammalian cells; SAXS data etc), following the feedbacks. We believe these changes have made the story more concise and focused.

Please find a detailed response to each of your comments below and we sincerely appreciate your time and contribution to improving the quality of our paper.

Reviewers' comments: (our response in blue)

Reviewer #1 (Remarks to the Author):

In this work, Pei et al. report a comprehensive biochemical characterization of a recently discovered T6SS effector, TseI from *Aeromonas dhakensis*. TseI is composed of a central RHS core, an N-terminal domain that interacts with VgrG and a C-terminal cytotoxic domain. Using a combination of site-directed mutagenesis, protein secretion assays, bacterial competition experiments, bacterial two-hybrid assays and pulldown assays, the authors addressed and answered several key questions that are crucial for the function of TseI. After identification of the active site of TseI and showing that it forms an immunity pair with TsiI, two autoproteolytic cleavage events were identified: both the N-terminal domain and the C-terminal domain are cleaved off from the large central RHS domain. While the cleavage of the C-terminus is conserved in other RHS toxins, the cleavage of the N-terminus is a unique feature of TseI. Next, the authors show that only the N-terminal domain and the toxic C-terminal domain are secreted, while the central RHS domain remains in *A. dhakensis*, and that secretion depends on the presence of both VgrG and TecI. Finally, the authors identified homologs of TseI in *Pseudomonas syringae* and *P. aeruginosa*, that both also show cleavage of the N-terminus and C-terminus.

The biochemical experiments were executed in an appropriate manner, and important controls were included. The work presented here describes the biological function of a T6SS-dependent RHS toxin and reveals some important novel findings, such as the cleavage of the N-terminal domain. I believe the article is suitable for publication in *Nature Communications*.

Thank you for the positive comments and detailed suggestions below.

However, there are some major issues that should be addressed before publication.

1. Introduction: A proper introduction to Tc toxins is missing. In particular the authors completely omit the work of the Raunser lab and do not cite it, although they use their atomic model of TccC3.

Thank you for pointing out this oversight. We have now added the following references where appropriate that should have been cited for the noted reasons.

1. Gatsogiannis, C. *et al.* Tc toxin activation requires unfolding and refolding of a α -propeller. *Nature* **563**, 209–233 (2018).
2. Roderer, D., Hofnagel, O., Benz, R. & Raunser, S. Structure of a Tc holotoxin pore provides insights into the translocation mechanism. **116**, (2019).
The encapsulated TcC C-terminal toxin was imaged in the structure of the Tc holotoxin in these two recent papers.
3. Meusch, D. *et al.* Mechanism of Tc toxin action revealed in molecular detail. *Nature* **508**, 61–65 (2014). *The structure template PDB 4O9X for TccC3 subunit in Photobacterium luminescens was reported in this study.*
4. Roderer, D. & Raunser, S. Tc Toxin Complexes: Assembly, Membrane Permeation, and Protein Translocation. *Annu. Rev. Microbiol.* **73**, 247–265 (2019). *A comprehensive updated review on*

Tc-toxin.

To make the story more focused, we believe a discussion of the Tc toxin C subunit with TseI and the teneurin extracellular domain is probably sufficient. We thus edited the following text in the discussion from line 288 “

Known structures of Rhs/YD-repeat homologs, including the bacterial Tc-toxin C-subunit TcC and the eukaryotic teneurin extracellular domain, show that the Rhs/YD-repeat core assembles a beta-barrel shell^{7,14,15,52–54}; the key structural difference between them is the position of the C-terminus (Supplemental Figure S10A). While the C-terminus of teneurin protrudes out of its shell^{52,53}, the Tc-toxin TcC shell and the TcB subunit encapsulate the C-terminal toxin that is released through the TcA-formed translocation channel when the holotoxin enters target cells^{7,15,55}. In comparison, expression of wild type and cleavage-defective TseI is toxic to cells suggesting the C-terminal toxin can reach its DNA substrate. Therefore, despite the predicted similarity of TseI to Tc-toxin TcC, TseI is unlikely to encapsulate its C-terminal VIRC nuclease toxin. This is further supported by the bacterial two-hybrid assay that showed direct interaction of non-cleaved TseI mutants, the protease-inactive D1407A-D1429A mutant and the protease deletion Δ RL40 mutant, with the immunity protein TsiI when they were fused to the split fragments of CyaA (Supplemental Figure S10B)....”

2. C-terminal toxin is not encapsulated: Since TseI is a homologue to TccC3 from *Phototribadus luminescens*, the cocoon or beta-barrel shell, as the authors name it, misses the B component, i.e. half of the barrel. Therefore, it is not surprising that the C-terminal toxin is not encapsulated. The SAXS experiment is vague, poorly done and illustrated and should be removed from the manuscript. The extension into which the C-terminus is modeled (Fig. 2i) could also be a part of the RHS cocoon or the N-terminal domain. The conclusion that the C-terminus is outside of the RHS cocoon is solely based on this SAXS experiment and should therefore also be removed from the manuscript. In addition, it does not make sense that the toxin is oriented in this direction. How can it remain bound to TseI after autoproteolysis if it is reaching out that far? This opens another important question that the authors can probably answer in a follow-up study: How can the C-terminal toxin remain attached to TseI after cleavage without an encapsulating barrel? A future structure of the protein would of course answer this question. In general, this study would greatly benefit from structural studies (such as X-ray crystallography or electron microscopy), that shed light onto the molecular organization of participating components, resulting in a more sophisticated model than currently presented in Fig. 7. We agree with the comments and, as also mentioned in editorial comments, we have removed the SAXS data from the manuscript.

Indeed, we are still puzzled how the cleaved N and C are still bound. We have attempted to use crystallography to solve the structures of wild type and mutant TseI proteins in the last two years by working with a protein structure expert. Based on our experience, TseI unfortunately belongs to one of those difficult proteins prone to aggregation/degradation and difficult for crystallization.

Another unaddressed question that could benefit from structural studies is the interaction between TseI with the chaperone TceI and carrier VgrG. We have recently started using cryo-EM and hope to get some structural insights. However, this is also a challenging task considering there hasn't been any chaperone structure reported yet and our multiple failed attempts for crystallization. In this current manuscript, we primarily focused on reporting the self-cleavage of T6SS Rhs effectors and their secretion using genetics and biochemical assays. But as the reviewer suggested, we will next work on gaining structural insights about the conformation of TseI alone and in complex with its binding partners in future work.

3. Discussion: “However, the Tc-toxin C-terminus has not been imaged inside the beta-barrel shell.” This is not true. The authors should read the literature in detail and cite the appropriate papers, accordingly. A good start would be: *Annu Rev Microbiol.* 2019 Sep 8;73:247-265

Again, we thank you for pointing out this oversight and have now updated the manuscript with this recent review as well as the research paper that published the holotoxin including the TcC C-terminus “*Gatsogiannis, C. et al. Tc toxin activation requires unfolding and refolding of a -propeller. Nature* 563, 209–233 (2018)” in appropriate places.

4. Figure legends do not provide enough detail.

We have revised all figure legends during this revision to include more details and in places where it is unfeasible, we have added a note referring to the text, methods, or supplemental supporting data.

5. Figure 7 is not conclusive. The authors should invest more time to develop a nice figure that should be properly described and discussed in the main text and in the figure legend.

We have now added a summary table of our findings accompanied with a protein complex model to capture the main conclusions.

6. Data presentation in general: The authors created a large number of TseI variants to address several particular features of TseI, such as cleavage of N- and C-terminus, toxic activity and export, by site-directed mutagenesis. This results in many different mutants whose effects are described by an identical set of experiments that are presented in several figures, therefore it is very difficult for the reader to follow. An overview figure in which all mutations and their respective location is shown would be helpful. In addition, it would also be helpful if one important experiment, such as the inter-species competition experiment (Fig. 1B ,D, Fig. 2D, Fig. 3B, Fig. 5B), would be presented as one concluding figure together with the mutant overview and the scheme in Fig. 7.

Thank you for this excellent suggestion. We have now added the relevant information in the new summary Fig. 6 to improve clarity in understanding phenotypes of different mutations. We believe the new summary table serves the same purpose of the competition figure.

In the following, I provide comments and suggestions to improve the paper in a point-by-point manner. There are some minor language issues that should also be changed.

Abstract:

Please write clearly that TseI is cleaved two times, releasing the C-terminal toxin and the N-terminal VIRN.

We have now revised the statement below: “*Here we demonstrate that a T6SS Rhs-family effector TseI in Aeromonas dhakensis is subject to self-cleavage at both the N- and the C-terminus releasing a middle Rhs core and two VgrG interacting domains which we name VIRN and VIRC (VgrG-interacting Rhs N/C-terminus).*”

Introduction:

p3, 15f.: Please re-write the sentence to make it clear that teneurins are only present in eukaryotes (animal kingdom).

We have added “eukaryotic” in the sentence below: “*One such example is the YD-repeat (tyrosine-aspartate) protein family whose members include many toxins in both gram-positive and gram-negative bacteria as well as eukaryotic teneurins that are conserved transmembrane adhesion proteins with critical functions in embryogenesis and neural development* ⁵⁻⁹”.

p.3, l.16f: I suggest to write bacteriophage-like tail instead of bacteriophage tail; and trans-membrane without the hyphen. If you decide to write “The tip is sharpened”, I suggest to write “sharpened” in “”. Thank you and we have revised the texts accordingly.

p.3, l21: Please explain what is meant with “structural effectors”. Modification of the cytoskeleton? Here “structural effectors” refer to evolved T6SS structural proteins, including Hcp, VgrG, PAAR, with extended functional domains. We have revised this sentence to clarify:

“Some effectors are Hcp, VgrG and PAAR structural proteins with evolved C-terminal functional domains...”

Old sentence

~~“Some Hcp, VgrG and PAAR proteins have evolved C-terminal functional domains and act as structural effectors”~~

p.4, l.4f: Please stress the double cleavage, like suggested for the abstract, and please describe clearly where and when cleavage occurs.

As suggested, we have revised the text below

“...possesses two self-cleavage sites, between residues C420 and P421 and between residues L1433 and S1434, resulting in three fragments, VIRN (VgrG-interacting Rhs N-terminus), Rhs core, and VIRG (VgrG-interacting Rhs C-terminus).”

p.4, l10f: Here you refer to TccC3 without a citation, and later in the manuscript (e.g. p.10, l.17) you refer to TccC3 and cite Busby et al., 2013, who describe the structure of the TcB-TcC subunit of Yen-Tc. The correct citation for the structure of TcdB2-TcCC3 (*Photobacterium luminescens*), pdb ID 4O9X, is Meusch et al., 2014 (doi: 10.1038/nature13015).

Thank you. We have corrected this reference as suggested.

Results:

p.4, l.19: Please explain your notation of *tsiI*.

We noticed this putative gene using a sequence management program Benchling that can auto-detect ORFs while processing imported DNA sequences. We have now added the sentence below and a reference for Benchling in the Methods section.

*“Benchling was used to manage and analyze DNA and protein sequences and to predict *tsiI* as a putative open reading frame.”*

p.5, l.3ff: It would be helpful to briefly explain the background of T6SS immunity pairs in this chapter. We have introduced effector-immunity pairs in the following sentence in the 2nd paragraph of Introduction section, line 41.

“Each antibacterial effector has a cognate immunity protein that confers self-protection...”

To avoid confusion and repetition, we now have added two relevant references

*“To test if *tseI* and *tsiI* encode a T6SS-dependent effector-immunity pair^{23,29},”*

p5., l.18f: “highly toxic”: Can you provide a comparison of toxicity with other T6SS effectors or other bacterial toxins?

Toxicity of T6SS effectors not only depends on their mode of action but also their synergies (LaCourse et al, 2018 PMID: 29459733), as well as the repairing response in different bacterial hosts where the effector is expressed or delivered into. The latter is demonstrated in our recent study (Liang et al, 2019, PMID: 31659021) in which a *Vibrio cholerae* effector TseL could kill effectively its immunity gene mutant but not *E. coli*. This is further discussed in a follow-up study on another effector TseH in which we show TseH is not sufficient to kill its immunity mutant or *E. coli* but can kill *Aeromonas* species

effectively due to innate protection pathways present in *E. coli* but not in *Aeromonas* (Hersch et al, 2020, Nature Microbiology, <https://www.nature.com/articles/s41564-020-0672-6>). In light of these studies, we believe comparing toxicity of effectors is context-dependent and not directly relevant here.

p.6, l.13: “secreted medium” does not make sense.

We agree and have deleted “secreted” wherever appropriate.

p.6, l.20f: Deletion of the whole Asp protease domain (39 residues) might result in a different orientation of the toxin domain. Can you exclude this possibility?

We agree that it is possible that removal of the whole protease domain could have affected the orientation/conformation. Since this refers to the bacterial two-hybrid assay that is now provided as a supplemental (new Supplemental Figure S10B), we didn’t pursue further except for the following two experiments. 1st, we show that deletion of this domain abolished its secretion (new Figure 3E); 2nd, purified mutant exhibited less stability than wild type (new Supplemental Figure S6F). Therefore, this protease deletion has additional effects in comparison with cleavage-defective point mutations.

p.7, l.7: Two recent publications show the encapsulation of the toxin domain in Tc toxins: Gatsogiannis et al, 2018 (doi: 10.1038/s41586-018-0556-6), Roderer et al, 2019 (doi: 10.1073/pnas.1909821116)

We have revised the text and included the above-mentioned references.

p.7, l.10ff: The Phyre models look like the two templates, indicating that TseI might look completely different. Therefore, these models are not really reliable, and I would show them in the SI and not as a main figure. I suggest to perform electron microscopy (negative staining of the purified protein is enough to get an idea about the shape) to get an additional proof that TseI looks like expected.

We agree with the suggestion and have moved the models from the main figures to supplemental figure S10A. We have indeed attempted to perform EM but didn’t obtain clear images at least partly due to protein aggregation. We are still in the process of solving this issue by optimizing conditions as well as using cryo-EM to resolve the TseI structure through collaboration. Since we don’t have the structural data yet, we have removed this from the results and added a brief discussion about this prediction based on the current genetic and biochemical evidence. The revised text can be found in line 295 and below “...despite the predicted similarity of TseI to Tc-toxin TcC, TseI is unlikely to encapsulate its C-terminal VIRC nuclease toxin. This is further supported by the bacterial two-hybrid assay that showed direct interaction of non-cleaved TseI mutants, the protease-inactive D1407A-D1429A mutant and the protease deletion Δ RL40 mutant, with the immunity protein TsiI when they were fused to the split fragments of CyaA (Supplemental Figure S10B). These results suggest the arrangement of Rhs-VIRC complex likely resembles that of the eukaryotic teneurin extracellular domain. This prediction warrants validation by structural characterization of TseI in future studies.”

p.7, l.25f.: The fact that secretion is not possible in trans, i.e. that it is only possible when the complex is pre-formed is an important finding. Does it mean that the complex made up of VIRN, the RHS-domains and the toxin domain cannot be reconstituted from the individual domains? Please provide this information.

The pull-down data of TseI different domains suggest the VIRN, Rhs, and VIRC can interact with each other. In addition, we show that TseI fragment remains in complex during protein purification regardless an N-His or a C-His tag is used. The chromosomal expressed TseI is also probably much lower than the plasmid-borne induced fragments. Therefore, this earlier observation is likely due to a number of factors. Since we have changed this section to compare the secretion of these plasmid-borne constructs between the *tseI* mutant and the T6SS null *vasK*, this statement has now been removed from the revised manuscript.

p.8, 1.21: Why should the secretion of the C-terminus be influenced by the N-terminal cleavage?
We have made substantial revision here by describing all mutations together. Therefore, the earlier statement is not applicable and thus removed.

~~“ Interestingly, TseIC secretion was not affected by the N-terminal cleavage.”~~

p.9, 1.13: What is meant with TseI-inactive mutants? Cleavage-inactive?
We meant nontoxic but this section of TseI effects in yeast cells has now been deleted.

p.10, 1.2ff: The pull-down results described here contradict the previous observations that TseI forms a non-covalent complex of all three parts. Can you provide an explanation? Does VgrG1 displace the VIRN and the RHS domain from the C-terminus of TseI?

We don't have evidence suggesting VgrG1 displaces VIRN and RHS especially VgrG1 interacts with VIRN. The lack of a full length TseI in those pull-down results is probably due to the efficient self-cleavage of TseI and detection was done with the antibody to the C-terminal 3V5 tag. We have consistently seen a much lower level of full length TseI in western blot analysis of cells expressing TseI and stained SDS-PAGE proteins gels with purified proteins (see new figures 2A, 2C etc).

p.10, 1.13: You introduce the rationale of the term VIRN domain here, but the expression is already used in the introduction. In the first paragraphs of the results section, you do not use VIRN, but refer to the domain as N-terminal. This can be confusing, therefore I recommend to introduce the notion VIRN in the introduction and to use it consistently throughout the manuscript. You furthermore showed that also the C-terminus interacts with VgrG1 (Fig.5D), which could then be called VIRC accordingly.

We thank you for this suggestion. We have now introduced VIRN and VIRC in the introduction and made it consistent thereafter wherever appropriate. However, in some figures we used the short form N and C for easy labeling.

p.11, 1.6ff: For PSPTO_5438, the toxic WT can apparently be expressed and purified (Fig.6D), whereas this is not easily possible for TseI. Please discuss why this is the case.

After learning TseI expression is very toxic, we expressed PSPTO_5438 together with its immunity protein as indicated in the legend and this helped us to purify PSPTO_5438 to sufficient quantity without killing expressing cells. The following text was described in the legend for new Figure 5D.
“N-terminal His-tagged PSPTO_5438 and mutant proteins were co-expressed with untagged immunity proteins, purified with nickel columns, and compared by SDS-PAGE analysis.”

Discussion:

p.11. 1.22: Teneurin citations are used to describe Tc toxins, please change (see above).

Thank you and we have changed the citation.

p.12., 1.7: To which inactive variants do you refer to (HFH-AAA)? With the toxic WT, I assume the cells die. What happens when the non-toxic variant is expressed (insoluble expression?) Please clarify. Yes, we meant the HFH-AAA nontoxic mutant. We have clarified this by changing “inactive” to “the nontoxic C-terminus mutant”. We could not purify the C-terminus fragment due to precipitation (new Supplemental Figure S7E).

p.12, 1.13: Please describe briefly what EsxAB is.

We have revised the statement as requested and added an additional reference:

“For example, the Mycobacterium T7SS secretes heterodimeric substrates EsxAB, of which EsxA displays membrane pore formation activities while its binding partner EsxB prevents EsxA aggregation^{50,51}”

p.13, 1.3ff: Is it only a common motif of TseI and protease self-cleavage or is there also cleavage site conservation or a common mechanism?

We think the original statement might be misleading. We didn't attempt to compare motifs of self-cleavage of TseI and proteases and Blast results suggest there is no common motif. To clarify, we have revised the statement below:

"TseI self-cleavage is also reminiscent of the common auto-processing of proteases..."

An additional section of the discussion, in which you compare the size of the RHS domain of TseI with those of Tc and Teneurins, would be informative. Is the TseI RHS domain possibly too small to encapsulate the C-terminus?

Thank you for the suggestion. We think this is probably not about the size based on the overlap of the predicted structural model of TseI with Tc-toxin TcC and teneurin shown in (Supplemental Figure S10A). In this revision, we tried to refrain from over-discussing the conformation and hope to address this definitively in follow-up structural studies.

Figures:

General remark: The figures are complex and almost every figure shows many different experiments. Therefore, it would be helpful for the reader to provide a brief description of the outcome of every experiment in the figure legend, especially in Figs. 2,3,5. In addition, a brief heading for every figure panel would help the reader to recognize which experiment is applied to address which point.

Thank you for these suggestions. We wrote the story and prepared the figures in the order of how the experiments were progressed. In the revision, we have restructured the sections to present what we found rather than how it was done. As a result, we have modified the figures extensively after taking careful consideration of all reviewers' suggestions and believe now the revised figures are clearer.

Fig. 1: It would be helpful to provide the residue numbering in the tseI scheme in Fig.1A, and to draw the respective domains in the real scale to each other. What is pfam15657, please write in the legend.

Agreed. We have now added the residue numbers and description for pfam15657 in the legend as follows *"Sequence of the VIRC toxin region was aligned with the consensus sequence of Pfam15657 that represents a conserved domain family of the predicted HNH/Endonuclease VII toxin with a characteristic conserved [ED]H motif and two histidine residues."*

Fig.2 C, E: In both cases, variant D1407 is shown. Is it the very same experiment and two times shown in two different contexts?

They are two different sample preparations. To make the story more concise, both N and C-terminal cleavage data have now been shown together in new Figure 2 and supplemental Figure S5, which have replaced the old figures.

Fig.1C and Fig. 2G: Is there a reason why the X- and Y-axis were changed in the two figure panels that show a bacterial-2-hybrid experiment?

It was simply due to that the experiments were done by two different students. We have redone this assay to make the format consistent.

Fig.2A,B: Is TseI-AAA in panel A the "parental" in panel B? Please name the variants consistently. Yes, and we have now specified mutations in figures wherever appropriate.

Fig. 2H,I: The Phyre-models look like the templates, therefore their significance is not very high. Which one was used in panel I? I suggest to put the phyre models to the SI and instead show a negative stain EM micrograph (or, more sophisticated, 2D classes of negatively stained TseI particles) that shows the shape of the molecule. To localize the N- and C-terminus of TseI in negative staining, immunogold labeling would be helpful.

We thank you for the suggestions. As explained earlier, TseI is challenging to handle in our hands due to protein aggregation. We have now removed the SAXS data and moved the Phyre models to Supplemental figure 10A. The predicted model of TseI using the Tc-toxin TcC as template was overlapped with the template TcC, and the predicted model based on teneurin was overlapped with the template teneurin.

Fig.3: Please provide a scheme of the variants MC and CT. In panel D, the non-specific signal that co-localizes with TseI-N is unfortunate and impairs the presence or absence of secreted protein, especially for pD435A. The context of the blot in panel E (T6SS-dependent secretion of the N-terminus) is difficult to understand, please provide a short description in the legend.

The scheme is now provided in Figure 6. Indeed, because of the non-specific signal shown in D, we wanted to confirm that TseI-N(VIRN) is secreted using an epitope-tagged TseI-N. This figure is now in supplemental Figure 6D with the following text added to the legend “

D, T6SS-dependent secretion of the VIRN domain of TseI. The VIRN was cloned to a pBAD vector with a C-terminal 3V5 tag and expressed in wild type, the $\Delta vasK$, and the $\Delta tseI$ strains.”

Is there a specific reason why variants E428A and E429A were used in different experiments?

No. E428A and E429A are phenotypically identical and we thus use them interchangeably for constructing combinational mutations.

In panel F, you refer to cleavage mutants, but you only show one mutant.

The new figures 3A and 3B replacing F now show the results of both N- and C- cleavage-defective mutants.

Fig.3 H: What should the right panel (anti-flag) show us? I can either see degradation of the in vitro expressed sample or unspecific binding.

For unknown reasons, the N-terminal anti-FLAG results show reproducible multiple bands in the *in vitro* expression assay. What we wanted to highlight in this figure is the detection of N+M and N+M+C in the E428A mutant and the E428A D1407A mutant but not the wild type, suggesting N-cleavage is subject to self-cleavage. In the revision, we now show a cropped region in main Figure 2D and moved the whole gel image to the uncropped figure file.

Fig.4: It would be helpful to provide the target organism as a heading to the respective figure panels.

For C, images of HEK cells treated with the different variants that potentially show differences in morphology would be informative (e.g. as a SI figure panel). What antibodies were used in B and C?

Thank you for the advice. Although we have removed the HEK data in the revision, we adopted this suggestion where appropriate by adding antibody information to the figures and legends.

Fig.5 A: Are vgrG1 and tecI deletion mutant strains? Please clarify.

Yes, they are. We have now added “deletion” mutant to the legend, as well as “ ” in the figure.

Fig. 5D: Why is the N fragment alone binding to VgrG, but not when full length TseI is provided? In this case, I would expect also binding of the N fragment after self-cleavage (or at least binding of the N-M fragment).

It is because TseI has its 3V5 epitope tag at its C-terminus and TseI is efficiently cleaved. All pull downs were performed using epitope-tagged TseI variants mainly because the customized antibodies to TseI were not available yet at that time. We have added the following statements in the legend of new Figure 4C to improve clarity

“Because C-terminal 3V5 tagged TseI was used in C to E, we could only detect the cleaved C-terminus but not full length TseI due to self-cleavage.”

We have also added the following text to line 205

“Because TseI was detected against its C-terminal 3V5 tag, we could only detect its cleaved C-terminus but not full length TseI probably due to efficient self-cleavage. As protein purification assay shows that the cleaved products remain in complex (Supplemental Figure S3B), we then tested the interaction of each TseI fragment with VgrG1 and TecI, respectively.”

Fig. 5E: A more detailed explanation would help the reader to understand why the C-terminal fragment is pulled down by TecI when full length TseI is provided as input.

The same reason as above. The added text should help clarify.

Fig.5 F: The experimental result suggests that the N-terminal domain directly interacts with the C-terminal domain without the central RHS domain. Please discuss this unexpected finding. Furthermore, the indication of “N” is missing.

We have revised following texts in the discussion line 274

“Although it remains unclear why TseI requires cleavage for its cell-to-cell competition, the VIRC toxin might require VIRN and Rhs core for stability at the physiological levels delivered to recipient cells. Importantly, we found that VIRN could interact with the VIRC toxin in the absence of Rhs, suggesting the chaperoning role of VIRN and Rhs could be synergistic.”

Fig. 5G: The signal for N- and C-fragment in the elution control due to unspecific binding is almost as strong as the actual signal in the other elutions.

We have replaced this with a clearer blot in new figure 4G.

Fig.6: Please indicate TseI clearly in A and B, and also the other three homologs that were expressed in this work. Please also provide the names of the organisms. In B, one of the Asp in “RhsA_Z5014” is not conserved, which should result in cleavage deficiency. If the authors also have access to this particular gene, it would be a great addition together with the other three homologs. Is there a particular reason why PSPTO_5438 is shown in Fig. 6, and VP1517 is shown in Fig. 8? It would be easier to read if both data were in one figure.

Thank you for the suggestion. We have highlighted TseI and other homologs used in this work in blue and added the organisms in A since A&B show the same proteins. We agree that RhsA_Z5014 doesn't have this conserved Asp which should affect its cleavage. Unfortunately, we don't have the strain or gDNA for cloning to test it here. Nonetheless, the other 3 homologs we tested clearly show cleavage is likely common among Rhs proteins.

We have now combined the data of PSPTO_5438 and VP1517 in the new Figure 5 and supplemental Figure S9.

Fig.7: This figure needs to be improved, both, on an illustrative and content-related view. The scheme should be more refined, both with respect to protein structures and cell shape. What is the reason to draw a direct interaction of N and C only in the target cell? Information from pulldown and secretion assays with TseI would help to establish a more complete picture of what is going on and could be ultimately implemented into Fig. 7.

We thank you for the suggestions and have made a new model figure in the revision. The role of immunity proteins in effector delivery remains elusive in the field especially for immunity proteins in the cytosol. As we don't have additional data on TsiI except for the interaction and protection, including TsiI would probably complicate the model. We hope the main findings are now presented more clearly in the revised schematic model in Figure 6.

Supplementary information:

SI Fig.2C: How does the alignment continue after the cleavage site? Please add more residues to visualize.

Homologous sequence predicted by Phyre2 ended there at the cleavage site. Sequence similarity of this region among homologs can be found in Figs 2B and 5B.

SI Fig.3: Do all fits of the SAXS data result in the same molecular shape presented in Fig. 2?

We have now removed the SAXS data.

SI Fig. 5: The legend refers to Fig. 3F, but no in vitro expression is shown in this panel.

The correct reference should have been old Fig 3H. We apologize for this error.

SI Fig. 6: Please provide the names of the organisms. The size of the respective RHS domains could also be provided (see my comment regarding the discussion above).

Thank you. We have now added organism names. Because the RHS domains have been highlighted with amino acid ruler shown above, the exact size for each RHS isn't shown to avoid overcrowding of the figure.

Reviewer #2 (Remarks to the Author):

In this manuscript, Pei et al characterize an *Aeromonas* T6SS effector/immunity pair named TseI/TsiI. The authors nicely describe a self-cleavage mechanism that releases the C-terminal toxin domain from the central Rhs region. They further discover an N-terminal domain named VIRN that is also cleaved off the Rhs, and is even secreted. It is suggested to function as an intramolecular chaperone. The concluding results suggest that these self-cleavage mechanisms are common in Rhs toxins.

This is an interesting study that will be of interest to the microbiology community, and especially to those interested in secreted bacterial toxins. There are some experiments that I think can be improved with additional controls to support the conclusions, and they are listed below. The section on possible activity of the effector in eukaryotic cells, however, is not properly supported by the experiments and should be either removed or improved.

Thank you for the positive comments. We have now rearranged the result sections to make the story more concise and removed the eukaryotic effects as suggested.

Major comments:

1) Much of this work relies on TseI secretion experiments, yet the authors did not show that the TseI secretion that they are detecting is indeed T6SS-dependent (to be clear, I am not questioning that TseI is a T6SS-delivered effector, as this was properly shown in competition assays). The authors only test

T6SS-dependent secretion for the N-terminal part in Fig. 3E. A secretion assay of TseI from WT vs. T6SS-mutant should be shown for the full-length protein (show the T6-mediated secretion of C- and N-terminal in the context of a full length effector).

We are very grateful for this suggestion. We have now added the *vasK* control for all TseI secretion assays in new Figure 3 and Figure 4.

2) Fig. 3A (p. 7 lines 22-23): Can the authors explain why the CT is secreted in *tseI*-deletion when expressed from a plasmid (far right lane), and also why is it not secreted then from WT? Since CT is expressed at a lower level in the cell compared to the full length TseI (as seen in the “Cell” lanes on the left), this secreted CT appears to be quite a significant amount. This result either contradicts the authors hypothesis, or it is the result of cell lysis/non-T6SS-dependent secretion (which refers back to my previous comment on the lack of T6SS-dependent secretion experiment).

We agree. Without the *vasK* control, we couldn't conclude whether the secretion was due to cell lysis or T6SS delivery. We performed the experiment after adding the *vasK* control and found that the previous weak signal of C-terminus toxin was probably due to cell lysis.

3) P.9 lines 4-6 (and Fig. 3F-G): since the CT was delivered even in the E429A mutant, but no complementation was seen during competition, the authors should show (immunoblot) that the mutant was expressed in the donor strain during the competition shown in 3F. Also, some explanation should be provided here (or in the discussion) about to the contradicting results shown in these panels. Is N-terminal cleavage required for prey intoxication/delivery or not? The question is left unaddressed. As suggested, we have now performed the western blot analysis using cells from the competition plates and added the data in new Supplemental Figure S6A, while the survival of prey and killer strains is shown in Figure 3A and Supplemental Figure S6B, respectively. The results confirmed that TseI and its variants were expressed.

Our new results using custom antibody to Rhs and N and the epitope V5 antibody show that E429A could be secreted in two fragments, the C-terminal toxin and the N+Rhs fragment (see Figure 3D). It is indeed puzzling that the E429A mutant can't kill in a cell-to-cell competition assay. We provided one explanation here in Figure 3F, in which we show that the N- and C-cleavage mutants were less toxic than wild type when expressed on pBAD vectors in *E. coli*. It is possible that the reduced toxicity at the much lower cell-to-cell delivery level might amplify this difference.

We have also added the following statement in line 193 to clarify:

“...The relative survival of cells between induced and uninduced conditions show that mutations D1407A and E429A attenuated toxicity in comparison with wild type TseI, although both TseI D1407A and E429A mutants also exhibited moderate toxicities in comparison with the nontoxic HFH-AAA mutant (Figure 3F). Considering that the physiological level of T6SS-delivered TseI is likely much lower than that of intracellular induction, such attenuated toxicity might account for the impaired killing of prey cell by cleavage mutants during competition.”

4) Fig. 4: The experiment shown in A is not necessarily testing for effector toxicity in yeast, but rather for stability of the plasmid. To directly show toxicity in yeast, the authors should clone the effector into a yeast inducible vector (e.g. under galactose-inducible promoter) and show that the expression of the effector is toxic. The current experiment possibly shows that the effector, being a DNase, is interfering with the transformation or with the stability of the plasmid (since the authors test for the presence of the plasmid that allows for growth in absence of uracil).

This seems to be a misunderstanding. We did clone TseI to a galactose-inducible yeast expression vector pRS416-Gal that was provided in the plasmid list. We had trouble when making this vector with wild type TseI with multiple failed attempts while the non-toxic TseI mutant was cloned without issue. This is likely due to leaky expression and TseI toxicity.

Given the weakness of current experimental setup that suffers from failed transformation, we have decided to remove all eukaryotic data in this revised manuscript.

5) To the best of my knowledge, no antibacterial T6SS DNase effector has been shown to perform as a trans-kingdom toxin. If the authors are making such a claim here, they need to provide further experimental support, such as visualizing effect on yeast/HEK293 DNA under a fluorescence microscope, or purifying DNA from cells expressing the effector and showing that it is degraded in vivo upon effector expression.

We thank the reviewer for the suggestions. The current bottleneck is transforming wild type *tseI* into eukaryotic cells. We have decided to study this in future studies and to focus this manuscript on T6SS-effector cleavage.

Minor comments:

1) P.5, lines 7-9 (Fig. 1C): Expression of TseI is toxic in bacteria (shown in Fig. 1D). How was it expressed in the BACTH strain without immunity (with VasH instead of TsiI)? Same question is relevant in Fig. 2G.

We used a catalytic mutant of TseI for the BACTH assays. To clarify, we have labeled the mutant H1497A in the new figure 1F and added the information in the legend of supplemental figure S10B.

2) It makes more sense that the detailed explanation on BACTH (p.6 line 24 – p.7 line 4) will appear when Fig. 1C is discussed.

We agree and have moved the explanation to accompany the first two-hybrid data in line 97.

3) p. 7 lines 1-12 (Fig. 2H): Since the N- and C-terminal domains were not modeled, this panel seems irrelevant and can be removed from the manuscript.

We have moved this to the Supplemental figure S10 as suggested by the 1st reviewer. This modelling could still be helpful to understand how the non-cleaved TseI could still be toxic and how its C-terminus could interact with the immunity protein.

4) p.8 lines 14-15: since now there are 2 cleavage sites, this sentence is confusing. I suggest to rephrase it so that the reader clearly understands which cleavage site is mentioned.

We have revised this section substantially by presenting both N and C-cleavage together and hope the new section is clearer now.

5) The numbers in the X-axis of Fig. 3C (Weblogo) are illegible.

We have corrected this.

6) Fig. 3D: please denote the non-specific signal from the N-terminal antibody with arrow or asterisk.

Thank you for the suggestion. We have now cropped images to only show the area of interest and moved the original images to the supplemental data or denoted non-specific signals in figures where needed.

7) Supplemental Fig. 4: while D345A retains N-terminal cleavage, it appears that all the tested

mutations were hampered in C-terminal cleavage, since the lower band seen in the TseIHFA-AAA lane is not present in any other lane. Can the authors comment on that?

This is due to that the control HFA-AAA is C-terminal His-tagged while the others are N-terminal His-tagged (new Supplemental Figure S5B). On the gel, the non-tagged C-terminus thus migrated faster (smaller) than the His-tagged C-terminus. The non-tagged C-terminus was detectable in the E429A and D435A mutants but not the others. This might be due to less total proteins loaded. However, this doesn't affect the conclusion drawn from experiment which clearly shows all purified mutants but the D435A defective in N-terminal cleavage.

8) Fig. 3D: can the authors comment as to why the secretion of TseIN is significantly hampered in the D435A mutant, even though N-terminal cleavage seems unaffected?

Based on the results in supplemental Figure S5B, S6E, we think D435A reduced the effectiveness of N-terminal cleavage rather than abolishing it.

9) Fig. 3F: why is a qualitative (n=1) spotting assay shown for competition now when all previous experiments were shown as quantitative (n=3) bar graphs? Was this not also performed in triplicates with 3 repetitions?

We have now presented this competition assay and included more mutants in new Figure 3A.

10) Fig. 3G: some controls are missing to show that the signal is T6SS-dependent. For example, show that no signal is detected when the pTseI plasmids are in a T6SS mutant.

We agree and have confirmed that the signal was due to cell lysis. We have now removed this assay in the revision.

11) Fig. 3H:

a) Why is E428A used instead of the E429A that was used in all previous assays?

There isn't any particular reason. Since E428A and E429A are phenotypically the same, we constructed mutants in parallel and this double mutant was tested first. This doesn't affect the conclusion that cleavage occurs similarly *in vitro* (new Figure 2D) and full length non-cleaved mutant TseI could be constructed as is shown in new supplemental Figure S6E.

b) It is not clear what is shown in the anti-FLAG panel. There are many bands and it is difficult to determine which band is relevant.

We wanted to show the upper bands of N+Rhs and full length for those mutants. The non-specific lower bands have been cropped out in the new figure 2D and the original is now provided in supplemental uncropped images.

c) It seems that in the E428A, the same pattern as in the parental lane is visible even though there should be no N-terminal cleavage (Supplemental Fig. 4). Can the authors please comment as to why they are seeing (what appears to be) N-terminal cleavage in the E428A mutant in this assay?

This should be a misunderstanding. E428 showed bands corresponding to N+Rhs (anti-FLAG), and full length and C only (anti-V5). To clarify, we have now added arrows and molecular weight to indicate these bands.

12) Fig. 5C (and P.10 lines 304): it would be helpful if the authors add size markers to this panel to allow the reader to see that only the TseI-C is being pulled down by VgrG1.

We agree and have added size markers to this gel as well as the other ones in this manuscript wherever appropriate. We could only detect TseI-C, which is probably due to the effective self-cleavage and the C-terminal His tag. In the following panels, we showed VgrG1 could interact with both the N and C-terminus of TseI. Note that the western blot was done using the C-terminal V5 antibody that can't detect the N-terminus or Rhs fragment so we can't rule out the possibility that VgrG1 pulled down the TseI complex and only C-3V5 was detectable. This doesn't affect our conclusion that VgrG1 interacts with the C and N-terminus of TseI shown in different panels of the new Figure 4.

13) Fig. 5G: in the elution panel, double clicking on the figure in adobe reader colors the figure in blue, and reveals significant bands detected even in the lane without His-M. Is this just the result of the reduced quality of the PDF figure?

This is probably due to the difficult nature of His-M that is prone to degradation. We have replaced the blot image with a clearer one in new Figure 4G.

14) P.10 lines 19-20: *Vibrio parahaemolyticus* RHS effector vp1517 had been previously predicted to be a T6SS effector (Salomon et al, Plos Pathogens, 2015).

We have now added this reference.

15) P. 12 lines 6-7: "In addition, the C-terminal toxin domain, active or inactive, could only be purified when co-expressed with the Rhs core and VIRN" – what figure does this refer to? Was this attempted directly?

We didn't show the data in the original submission. When we tried to make TseI custom antibodies, we could purify N and M only fragments but couldn't express/purify C-only fragment. We have now added this new Supplemental Figure S7E in which C was barely seen in induced cell lysates and elution samples.

16) P. 12 lines 15-16: "... the toxin might require VIRN for stability when delivered to recipient cells..." - since the possible function of VIRN as a chaperone is one of the central conclusions in this work, perhaps the authors should test expression of the C-terminal toxin domain in *E. coli* to determine if it is stable and toxic on its own, and if co-expression of VIRN affects the stability or activity of the toxin domain? That would explain why VIRN is also secreted. On the same topic, the model in Fig. 7 suggests that N and C interact after delivery to the prey cell, but no experimental support for this is provided.

Expression of C is toxic but not as toxic as the wild type (new supplemental figure S7C). After taking all the reviewer comments, we have revised our schematic model in Fig 6 that summarizes our main findings and a simple model of TseI complex with the VgrG and chaperone.

Reviewer #3 (Remarks to the Author):

Intramolecular chaperone-mediated secretion of Rhs/YD-repeat effector toxin by the type VI secretion system

Comments to the authors:

In this paper the authors investigate the function and delivery mechanism of an Rhs homolog from *Aeromonas dhakensis*. The authors find that this effector (known as TseI) is mobilized by T6S and delivered to other *Aeromonas dhakensis* bacteria where it inhibits the growth of bacteria lacking cognate immunity. The toxic effect of the C-terminal toxin is confirmed to be a DNase and an adjacent gene provides immunity by direct binding to the C-terminal toxin. The authors further characterize the protein and find that it is subjected to autocleavage (similar to secreted Rhs toxins (Tc toxins) in the C-terminus as well as the N-terminus. The authors show that the three fragments remain bound to one another by immunoprecipitation. Surprisingly, they show that the C-terminal toxin is not encapsulated within the Rhs-core as suggested previously, but that it is cleaved off and attached to the outside of the Rhs-core. The Rhs-core is not secreted but remains in the delivering cell, whereas the N-terminus and C-terminal toxin are secreted. The authors propose a model where the Rhs-core, the N-terminal fragment of TseI and TEC acts as chaperons that facilitate delivery of the TseI C-terminus to neighboring cells. If correct, this is a very interesting and previously unrecognized aspect of Rhs protein homologs and would be very interesting to the field. However, significant amount of detail is lacking from the manuscript. For example, there are no descriptions of how the plasmids in the study were generated, or what parts of the protein antibodies were raised against. No raw data is provided for the figures and a number of important controls are lacking. This raises a number of concerns listed below which need to be addressed in order to make this manuscript suitable for publication.

Thank you for the constructive comments below.

Main concerns:

1) The conclusion that the toxin must be exposed within the cell based on the fact that transformation of a plasmid with TseI results in toxicity in bacteria, yeast and HEC283T cells without immunity. This is not the only reason that these constructs could give toxicity. There could be internal expression of the C-terminal toxin from an internal translation start site within the C-terminus that results in toxicity. The same is true for the yeast two hybrid screen. Thus, the toxicity data do not prove that the C-terminus is exposed and toxic while bound to the Rhs-core. The authors would need to check if there is internal expression of their toxin by for example introducing stop codons within the TseI ORF before the DPXGL motif in order to show that the C-terminus is only toxic when expressed from the full-length ORF.

There might be some misunderstanding. We had considered the possibility of an internal translational start site as the reviewer suggested but ruled it out because the C-terminal cleavage-defective D1407A and D1429A mutations show no such product in all the relevant western blotting assays.

2) The authors show that the non-cleavable E429A still secretes C-terminal toxin, but is unable to inhibit target cells. Yet, in Fig 3G it seems as the toxin is indeed delivered to target cells. This is very contradictory and should be commented upon, how do the authors suggest this would work mechanistically? Why would the toxin not be toxic in the receiving cell? However, many controls are missing for these experiments (listed below), which might clarify the picture.

Thank you for asking these important questions. We believed it was because both N and C are required to cause toxicity at the physiological condition, ie delivered by T6SS rather than plasmid-borne expression. We have added a section in this revision (line 189-198) to show that cleavage is important

for toxicity since non-cleaved N or C mutants exhibited reduced toxicity in comparison with wild type (new Figure 3F).

a. Fig 3G. The HiBiT/LgBiT assay is not convincing. How do you know that LgBiT is actually delivered to the cells and not to the extracellular milieu? The color could come from lysing targets releasing HiBiT to the supernatant. Since you have a functioning split cAMP assay, you could repeat the experiment with these strains and show that actual intact individual cells receive the toxin. Or, LgBiT fused to a protein known to be secreted to the extra-cellular milieu could work as a negative control. The pMC control is not valid as it is not delivered outside of the cell. At a minimum a control showing the lack of cell lysis in the assay should be included.

Thank you. After adding supernatant and pellet controls, we found that both the supernatant and cells could show luminescence signals suggesting the complementation of HiBiT/LgBiT could occur in the supernatant from lysed cells. This might be due to the high sensitivity of this enzymatic assay. In the revision, we have deleted this assay.

b. Fig 3D. An appropriate control i.e. a protein that is secreted should be added to the analysis. We have now added Hcp, a hall mark for T6SS secretion, as a control in all secretion assays where appropriate.

c. Fig 3D. What is the extra band in the N-terminal blot? This is not present in Figure S9 where the antibodies are verified, could you please expand on this? Why is the Rhs core so much less abundant than the NT and CT? If these are indeed cleavage products, would you not expect a ratio of 1:1:1. Could this be the reason why it is not detected in the secreted fraction? Also, the mutants seem to make even less Rhs core, could this be the reason that there is less inhibition?

Purified TseI proteins and SSU cell lysates were used to validate the antibodies. In old Figure 3D, *A. dhakensis* whole cells were used which explains the multiple non-specific signals. This is consistent with the validation results using whole cell lysates. Because the nonspecific band was also detectable in the secretion samples, we used an epitope-tagged TseI-N to confirm this secretion result shown in new Supplemental Figure S6D.

d. Fig 3E Cfu values for the inhibitor should be shown in addition to the target. Otherwise it isn't possible to assess how expression of the different TseI variants affect inhibitor fitness. If inhibitor fitness is affected, this could be the reason that you do not see any inhibition in the competition. I believe the reviewer is referring to Figure 3F. We have now added killer survival of the competition assay for all the competition assays in this revision except for Figure 1B which shows the immunity gene complementation rescues survival.

3) Over-expression in the eukaryotic cells to show toxicity does not really add to the paper, unless you could also show that the toxin is actually delivered to these cells. Could you do competitions with yeast/HEC 283T cells?

We agree and have now deleted all eukaryotic toxicity test as suggested by reviewer 2. *A. dhakensis* T6SS can't kill yeast cells probably due to penetration deficiency. It possesses other toxins including T3SS effectors and aerolysin that may target mammalian cells.

4) Although it is likely that the protein is a nuclease considering the HNH nuclease domain, more evidence or controls are needed to prove this. The loading lane has been cut from the gel in Fig 1A and

there are smears in the TseI samples. It is possible that TseI binds to the DNA and gets stuck in the loading lane, therefore it is important to show the lane. The samples could also be run on a protein gel and stained for DNA. Or, actual DNase activity could be shown in bacteriophage over-expression or receiving toxin by DNA staining and microscopy.

We have now shown the full gel picture with loading lane visible that doesn't show any trapped DNA. We believe the in vitro assay is sufficient for this activity test to show nuclease activity. We thank the reviewer for suggesting the additional experiments though.

5) The secreted fraction on western blots in figures 2E and 3D are not very clean. In figures 2E and 3D in particular, other components of the TseI protein (except the C-terminus) can be identified in the secreted fraction. Yet, the authors claim that only the C-terminus and N-terminus are secreted. In addition, an appropriate control i.e. a protein that is secreted should be added to the analysis.

Thank you for this important comment. We have now compared detection of TseI using epitope antibodies and protein-specific antibodies and found that full length TseI and the middle Rhs core could also be secreted when using the antibody to TseI but not the epitope tags. We have now revised the figures and models accordingly.

We have added the secretion of Hcp as a positive control for secreted proteins in all assays where applicable.

6) The authors claim that mutation of D1407A and D1429A do not affect N-terminal cleavage, but on the SDS gel it is obvious that the N-terminal fragment in these mutants is less abundant (Fig 2B). In addition, two additional larger bands appear, which look as they would correspond to full-length and full-length – NT, where most of the protein seems to be full-length though. Could the authors comment on this?

Thank you for the comments. We agree and have added the following text in line 144 " *Notably, although N-terminal cleavage is still detectable in the C-terminal cleavage-defective mutants, the amount of full length TseI was substantially enriched in comparison with wild type (Figure 2C&D, Supplemental Figure 5A&C), suggesting that the C-terminal protease-inactivating mutations reduced the efficiency of N-terminal cleavage.*"

7) A full annotated genome of the *Aeromonas* strain is not available on NCBI, only shotgun sequencing. This makes it difficult to evaluate the genetic mutations created in the study. Could the authors please provide accession no:s of the contigs, gene ID:s for the genes relevant in this study.

The genome assembly we used as a reference in this study is GenBank NZ_JH815591.1 (https://www.ncbi.nlm.nih.gov/nuccore/NZ_JH815591.1). To help readers find the relevant gene sequences, we have now also provided the gene sequences of the hcp1-tsiI 5-gene cluster in fasta format as supplemental dataset 3. We have added this statement to the methods

"All gene sequences of *A. dhakensis* SSU are retrieved from the draft genome assembly (GenBank NZ_JH815591.1). The gene sequences of *hcp1-vgrG1-tecI-tseI-tsiI* are provided in Supplemental dataset 3."

8) Raw data is not provided for the experiments. Please provide raw data for all experiments.

We have now provided the raw data for cropped western blot images and killing assays.

9) Cfu values for the inhibitor should be shown in addition to the target. Alternatively, immunity protection controls for each competition could suffice as well. Otherwise it isn't possible to assess how

expression of the different TseI variants or mutations of the T6S components affect inhibitor fitness. If inhibitor fitness is affected, this could be the reason that you do not see any inhibition in some competitions.

We have now added the survival CFU of inhibitors in all killing assays wherever appropriate.

10) From fig s7 it seems as the effectors lack PAAR domains? If so, this should be commented on in the text.

Yes, not all TseI homologs have PAAR domains. We have made a comment in the results page, line 239

“The N-terminus sequences show variable lengths, with some possessing the T6SS-associated PAAR motif”

Figure 5. There is a large inconsistency in the background seen when using the anti-V5 antibody. In some of the blots like 5D and 5E there are almost no unspecific binding and in others like 5F, the background is massive. Why is that?

We used the same commercial antibody from Thermo. The background in the whole cell samples might result from different amount of proteins loaded or exposure time. The Rhs core 3V5-M generates the most unspecific signals especially in high abundance. This instability may have contributed to our failed crystallography attempts. The background in the eluted samples seems to be OK probably due to low abundance.

Figure 4. The authors use a transformation assay into yeast/HEC293T cells to say that TseI is toxic also in eukaryotic cells. However, there is no control for the transformation of wild type TseI, only negative data. It could just be that this plasmid was not taken up by the yeast/HEC293T cells. A regulatable promoter should be used instead so that the same transformation can be plated/grown on inducing /repressing conditions in order to prove that it is expression per se that is toxic.

Thank you for the suggestion. As explained to the other comments, we did use a galactose-inducible yeast-expression plasmid and couldn't obtain viable transformants while plasmids carrying nontoxic *tseI* mutations were readily transformed into yeast cells. We have now removed this toxicity assay in eukaryotic cells.

Minor comments:

1. Page 5, line 13. “Expression of TseI and its mutants, H1497A and HFH-AAA, in *E. coli* showed that survival of cells was reduced by 5-logs when expressing TseI in comparison with cells expressing *tseI* mutants (Figure 1D), indicating that the mutations abolish toxicity.” Unclear and indirect, please rephrase.

We have replaced it with the following text in line 82.

*“...Wild type TseI and its two mutants were expressed using an arabinose-inducible vector pBAD in *E. coli*⁴³. Survival of *E. coli* was severely reduced in wild type samples compared with the mutants, indicating that H1497A and HFH-AAA abolished TseI toxicity (Figure 1C).”*

2. Page 6, line 12. “We found that only wild type TseI C-terminal fragment was detected in the secreted medium suggesting full length wild type TseI and its two non-cleavage mutants D1407A and D1429A are not secreted (Figure 2C).” The sentence lacks a comma after medium and that after suggesting. In addition, TseI and its mutants is not grammatically correct, please rephrase to: “wild type and mutant TseI” throughout the paper. This is also a very bold statement considering that there are additional bands in many of the blots (see above).

As suggested, we have rephrased to “*wild type and mutant TseI*” throughout the paper wherever appropriate. As explained earlier, we have performed additional

3. Page 8 Line 19. Please give aa numbers to the EED residues mutated.

This statement has been removed from the revision but the EED residue numbers have been added to the legend of Supplemental Figure S5.

4. Page 8, line 20. Do you mean Fig S5 instead of or in addition to S4?

Actually, we believe it was correct referring to S4.

5. Add size estimations for the westernblots in Figs 3D, 3E, 3H5B-G. Also, when blots are cropped for the main figures the entire gels should be shown in supplement.

We have now provided size markers and uncropped images in this revision.

6. Figure 4B-C. What antibody was used to detect TseI in yeast and HEC293T cells?

Anti-V5 antibody was used to detect C-terminal 3V5 tagged TseI under those conditions.

7. Figure 4B. Since the D1407A mutation results in Rhs-core + CT and NT fragments and not full-length TseI, it should say “core” on the left of the blot and no “full” as this is confusing.

This figure has been removed in this revision.

8. Figure 4C. Can full length TseI be detected in the HEC293T cells? If not, is it possible that only the C-terminal toxin is produced from another promoter within the ORF?

Unfortunately we could only detect nontoxic TseI mutant but not the wild type. This figure has been removed from the revision.

9. Figure S2. Why do you get less of the N-terminal fragment when purifying with an N-terminal his-tag?

The N-terminal tag might have affected stability or column-binding. This is supported by the poor purification using both N and C-terminal his-tag in the 3rd sample.

10. Figure legends are not adequate in explaining what the figure shows. Remove all the interpretation of the data from the figure legend (e.g Figure S4 and S8). The authors should be allowed to interpret the data themselves. Please provide more details about what is shown in the figure instead. See examples of lacking detail below (this is not a comprehensive list).

We have now revised the paper following the suggestions below throughout to make it clearer for all legends.

a. All figures: What organisms were used for the different experiments? Sometimes this is stated, sometimes not.

b. Figure 1D and 2F. What dilutions were plated?

c. Figure 2B. Purified proteins – how were the proteins purified?

d. Figure 4B-C. What antibodies were used?

e. Figure S4 and S7. What does 40mM and 250mM stand for?

f. Figure S9. Where is the data with the *tseI* mutant?

REVIEWERS' COMMENTS:

Reviewer #1 (Remarks to the Author):

The authors have appropriately addressed all points raised. From my side, the manuscript is ready to be published. Congrats to a nice study!

Reviewer #2 (Remarks to the Author):

The authors properly address previously raised concerns in this revised version.

I do wish to note a few minor comments to consider regarding some of the revised figures:

1) The numbers below the weblogs in Fig. 2 are still too small, and they also don't match the larger numbers denoting the mutations (for example, D1407A points at 1626). Also, I can still see the original bit numbers behind the overlaid numbers (Y axis) when zooming in.

2) Supplemental Figure 4 – the resolution of the image is very low, and the text in the outer rim is illegible. Is it supposed to be strain names? Please provide a higher resolution figure.

3) Figure 2D – the authors neglect to address the fact that this figure does not show the N-terminal part in the parental and D1407A lanes. This should somehow be described in the text.

4) Figure 3C-D and 4H: The correct vasK deletion background here should have been a deletion of tseI (delta-tseI/delta-vasK). Using a vasK deletion in which endogenous TseI is still expressed is somewhat misleading when it is compared to a delta-tseI strain. As shown in Supplemental Figure 6D, the endogenous TseI may compete with and reduce the secretion levels of ectopically expressed TseI fragments (less VIRN is secreted from WT compared to tseI deletion background).

Reviewer #3 (Remarks to the Author):

Intramolecular chaperone-mediated secretion of Rhs/YD-repeat effector toxin by the type VI secretion system

Comments to the authors:

In the revised manuscript by Pei et al the authors have addressed most of my previous comments to satisfaction. Overall, the revised manuscript is improved and should be of interest to, and comprehensible for, a wider audience. Some minor comments remain to be addressed however:

1. In response to my first main concern that the data presented does not support the conclusion that the toxin must be exposed rather than being hidden within the Rhs shell, the authors reply that: "There might be some misunderstanding. We had considered the possibility of an internal translational start site as the reviewer suggested but ruled it out because the C-terminal cleavage-defective D1407A and D1429A mutations show no such product in all the relevant western blotting assays." As the western blots were performed with strains over-expressing the TseI protein from a strong T7 promoter, it is very likely that the low level, native expression that would be expected from the internal promoter and suboptimal start codon would not be detected on the same western

blot. Based on this data, it is therefore not possible to determine whether the toxicity of the uncleavable mutants is a result of internal expression of the toxin or from the toxin being exposed. This should be commented on in the discussion. For reference of internal expression see: Starsta et al. PLOS genetics 2020. <https://doi.org/10.1371/journal.pgen.1008607>

2. Legend of figure 3, line 367 "from arginine 1394 to lysine 1433" should say "from arginine 1394 to leucine 1433".

REVIEWERS' COMMENTS:

Reviewer #1 (Remarks to the Author):

The authors have appropriately addressed all points raised. From my side, the manuscript is ready to be published. Congrats to a nice study!

Thank you!

Reviewer #2 (Remarks to the Author):

The authors properly address previously raised concerns in this revised version.

Thank you!

I do wish to note a few minor comments to consider regarding some of the revised figures:

1) The numbers below the weblogos in Fig. 2 are still too small, and they also don't match the larger numbers denoting the mutations (for example, D1407A points at 1626). Also, I can still see the original bit numbers behind the overlaid numbers (Y axis) when zooming in.

Thank you for pointing this out. As stated in the legend, the weblogo was generated using the sequence alignment of 48 representative Rhs homologs. Therefore, the small numbers below the weblogos correspond to the positions in the alignment file (with gaps and insertions) but not the amino acid positions of the TseI protein. To avoid such confusion, we have removed the small numbers and only highlighted the key residues of TseI relevant to this paper. We have carefully checked the new image to make sure it is now clean and not showing any overlaid numbers.

2) Supplemental Figure 4 – the resolution of the image is very low, and the text in the outer rim is illegible. Is it supposed to be strain names? Please provide a higher resolution figure.

A full vector image is now provided and strain names are legible with magnification.

3) Figure 2D – the authors neglect to address the fact that this figure does not show the N-terminal part in the parental and D1407A lanes. This should somehow be described in the text.

We have now added the following statement below to line 193 “

Using the anti-FLAG antibody, we did not detect a distinct band corresponding to the cleaved N-terminus or full length TseI in parental and D1407A samples. Instead, we found multiple nonspecific signals across all samples which likely result from lower stability of TseI N-terminus under the in vitro expression condition (Figure 2 C and D, and source data).”

4) Figure 3C-D and 4H: The correct *vasK* deletion background here should have been a deletion of *tseI* (*delta-tseI/delta-vasK*). Using a *vasK* deletion in which endogenous TseI is still expressed is somewhat misleading when it is compared to a *delta-tseI* strain. As shown in Supplemental Figure 6D, the endogenous TseI may compete with and reduce the secretion levels of ectopically expressed TseI fragments (less VIRN is secreted from WT compared to *tseI* deletion background).

We understand the reviewer's point that "the endogenous TseI may compete with and reduce the secretion levels of ectopically expressed TseI fragments", but we believe the *vasK* mutant can serve as a sufficient control to support the conclusion for the following reasons.

VasK is a key structural component of T6SS, and the *vasK* mutant has been used as a T6SS-inactive control in numerous studies. Here we also confirmed that Hcp, the most abundant T6SS-secreted protein, is not secreted in the *vasK* mutant under the conditions tested.

These figures (Figure 3C-D and 4H) show that TseI mutants and fragments were secreted in the *tseI* mutant complemented with plasmid-borne TseI constructs. To confirm that the observed secretion is T6SS-dependent, we expressed these same plasmid constructs in the T6SS null *vasK* mutant and found no secretion.

If the no-secretion was due to competition from endogenous TseI, we would expect to detect the endogenous TseI in the secretion samples of the *vasK* mutant, which is not the case. In addition, the endogenous TseI would likely to be present at much lower levels than the arabinose-induced ectopically expressed TseI.

Reviewer #3 (Remarks to the Author):

Intramolecular chaperone-mediated secretion of Rhs/YD-repeat effector toxin by the type VI secretion system

Comments to the authors:

In the revised manuscript by Pei et al the authors have addressed most of my previous comments to satisfaction. Overall, the revised manuscript is improved and should be of interest to, and comprehensible for, a wider audience.

Thank you.

Some minor comments remain to be addressed however:

1. In response to my first main concern that the data presented does not support the conclusion that the toxin must be exposed rather than being hidden within the Rhs shell, the authors reply that: "There might be some misunderstanding. We had considered the possibility of an internal translational start site as the reviewer suggested but ruled it out because the C-terminal cleavage-defective D1407A and

D1429A mutations show no such product in all the relevant western blotting assays.” As the western blots were performed with strains over-expressing the TseI protein from a strong T7 promoter, it is very likely that the low level, native expression that would be expected from the internal promoter and suboptimal start codon would not be detected on the same western blot. Based on this data, it is therefore not possible to determine whether the toxicity of the uncleavable mutants is a result of internal expression of the toxin or from the toxin being exposed. This should be commented on in the discussion. For reference of internal expression see: Starsta et al. PLOS genetics 2020. <https://doi.org/10.1371/journal.pgen.1008607>

We agree that our current data cannot determine the existence or non-existence of an internal promoter or suboptimal start codon effect and thus revised the statement from “...TseI is unlikely to encapsulate its C-terminal VIRC nuclease toxin...” to “... TseI might not encapsulate its C-terminal VIRC nuclease toxin...” in the discussion.

When the non-cleaved D1407A or D1429A *tseI* is induced by the T7 or the pBAD promoter, there should be abundant mRNA transcripts regardless of the hypothetical existence of any internal promoter. If there is any internal ribosomal binding and translational starting site, its translation would be very inefficient since we didn't see any products that would suggest its existence.

Thank you for providing the reference, in which internal promoters were found in the *rhs* genes in *Salmonella* Typhimurium to drive the expression of the toxin domain. Note that the full-length *rhs* genes were not expressed in *Salmonella* Typhimurium, which is not the case in our study. Given that TseI is expressed well, the possibility of selecting for an internal and much less efficient transcriptional/translational product would be low. Internal transcription/translation is interesting but likely beyond the scope of this study.

Ultimately, as we stated in the discussion, structural characterization is needed to definitively determine the arrangement of the Rhs-VIRC toxin.

2. Legend of figure 3, line 367 “from arginine 1394 to lysine 1433” should say “from arginine 1394 to leucine 1433”.

Thank you very much for catching this and we have made the correction accordingly.